# Prasugrel inhibits TLR7-driven autoimmunity in systemic lupus erythematosus by acetylating cGAS

Zeng-Lin Guo[1,7], Li-Ming Sun [1,7], Shuai Jiang[1,7], Ming Zhao [1,7], Yuhui Li[2], Jinjing Qian[3], Yakai Fu[3], Chunmei Wu[3], Ying Yuan[1,4], Wen Xue[1], Shao-Zhen Jiang[1,5], Sen-Chao Yuan[1], Xucheng Lv [6], Xingxing Yang[6], Lehua Yin[1], Peng-Peng Zhu[1,4], Yu Yu [1], Xin Xu[1], Kai Wang[1], Qiu-Ying Han[1], Zhuoxin Li [1], Zhi-Hui Su[1], Xi-Ping Yu[1], Jiaqi Wu [1], Hong Cai[1], Tian Xia[1], Yuan Chen[1], Xue-Min Zhang [1,4,5], Wei-Hua Li[1], Ai-Ling Li[1,4,5], Tao Zhou [1,4], Zhanguo Li [2], Qiong Fu [3] ✉, Xinhua He [1,6] ✉ & Tao Li [1,4,5] ✉

Systemic lupus erythematosus (SLE) has a complex, multifactorial etiology, which contributes to a lack of definitive cure and limited treatment efficacy. Here, we report that cyclic GMP-AMP synthase (cGAS) is significantly activated in SLE patients. We further demonstrate that cGAS deletion protects mice from lupus-like symptoms induced by the TLR7 agonist imiquimod (IMQ). In a screen of 3,159 FDA-approved drugs, we identify the antiplatelet agent prasugrel as a potent cGAS inhibitor. Mechanistically, prasugrel disrupts the DNA-triggered liquid phase condensation and activation of cGAS via direct acetylation. Strikingly, we find that prasugrel exhibits remarkable efficacy in treating SLE in both mouse models and patient cells. Importantly, we report elevated plasma cyclic GMP-AMP (cGAMP) in SLE patients and identify it as a potential biomarker for predicting prasugrel response. Thus, our work elucidates the essential role of cGAS in SLE pathogenesis and presents prasugrel as a promising therapeutic option with immediate translational potential.

Systemic lupus erythematosus (SLE) is a complex autoimmune disease characterized by the production of autoantibodies and the activation of immune cells that can attack a variety of tissues and organs throughout the body[1]. The pathogenesis of SLE is complicated and its clinical manifestations are heterogeneous, posing a formidable challenge in finding a definitive cure for SLE[2,3]. The existing treatment approaches typically focus on addressing individual pathological conditions and aim to alleviate symptoms[2,3]. However, these treatments lack precise targets, leading to various side effects and limited efficacy[3]. Studies revealed that the dysregulation of nucleic acid tolerance can lead to lupus-like diseases[4,5]. It is also reported that the transcriptional features closely associated with type I interferon (IFN) and the production of various autoantibodies, especially anti-double-stranded DNA antibodies and anti-nuclear antibodies, are prominent hallmarks of SLE[2,3]. Therefore, the involvement of the key DNA sensor, cyclic GMP-AMP synthase (cGAS), in SLE is of great interest[6].

By sensing cytosolic DNA, cGAS plays a critical role in immune responses against infections[7]. The emergence of DNA in the cytoplasm is normally an important danger signal for pathogen invasion[8]. cGAS detects cytoplasmic DNA and catalyzes the synthesis of 2'3'-cyclic

[1]Nanhu Laboratory, National Center of Biomedical Analysis, Beijing, China. [2]Department of Rheumatology and Immunology, Peking University People's Hospital, Beijing, China. [3]Department of Rheumatology, Renji Hospital, Shanghai Jiao Tong University School of Medicine, Shanghai, China. [4]Institute of Translational Medicine, Zhejiang University, Hangzhou, China. [5]School of Basic Medical Sciences, Fudan University, Shanghai, China. [6]Institute of Pharmacology and Toxicology, Beijing, China. [7]These authors contributed equally: Zeng-Lin Guo, Li-Ming Sun, Shuai Jiang, Ming Zhao. ✉e-mail: fuqiong@renji.com; hexinhua01@126.com; tli@ncba.ac.cn

GMP-AMP (cGAMP), a second messenger molecule[9,10]. 2'3'-cGAMP then binds to the stimulator of interferon genes (STING, also known as MITA, MPYS or ERIS), an endoplasmic reticulum-localized adaptor protein[9-14], mediating the production of type I IFNs and a spectrum of proinflammatory cytokines to elicit anti-infection immunity.

Besides the pathogen infection-introduced DNA, self-DNA can also stimulate cGAS activation[4,15-19]. For example, the release of mtDNA triggers immune responses by activating cGAS, and this process is implicated in multiple diseases[4,17,18]. In addition, the three-prime repair exonuclease 1 (TREX1) is responsible for metabolizing self-DNA fragments[20-22]. The deficiency of *TREX1* has been identified in patients with SLE[23] and Aicardi-Goutières syndrome (AGS)[24-26], and the *Trex1*[-/-] mice exhibit similar symptoms seen in human AGS patients[26]. The deletion of *Cgas* in *Trex1*[-/-] mice effectively ameliorates the disease manifestations, demonstrating that TREX1 deficiency leads to cytosolic DNA accumulation that stimulates autoimmune responses via cGAS[16,19,27]. Similarly, it is postulated that cGAS may be a key driver for SLE pathogenesis because of the type I IFN-associated transcriptional profiles and the aberrant nucleic acid-metabolizing processes found in SLE[1,5]. Despite this association, a recent study reported that cGAS did not promote autoimmunity in both pristane-induced murine SLE model and the MRL/*lpr* model[28], suggesting that the specific mechanisms by which cGAS influences SLE, and its potential as a therapeutic target, remain obscure. Therefore, investigating the role of cGAS in SLE using clinical samples is of significant importance to elucidate the mechanisms and assess the therapeutic potential of cGAS.

In the current study, we find that cGAS is significantly activated in SLE patients and cGAS deletion protects mice from lupus-like symptoms induced by TLR7 agonist, imiquimod (IMQ). We further identify prasugrel, a widely used antiplatelet agent for blood clot prevention, as a potent cGAS inhibitor. Prasugrel directly acetylates cGAS and disrupts DNA-triggered liquid phase condensation, thereby blocking cGAS activation. Importantly, prasugrel exhibits remarkable efficacy in treating SLE in both IMQ-induced mouse models and patient cells. Our study elucidates the critical role of cGAS activation in SLE etiology and uncovers a novel therapeutic application for prasugrel.

## Results

### cGAS mediates the autoimmunity in TLR7-driven systemic lupus erythematosus

To investigate the involvement of cGAS in the pathogenesis of SLE, we measured the plasma concentrations of cGAMP, the product synthesized by the activated cGAS[9,10], in samples obtained from healthy donors and SLE patients. Consistent with a previous study[29], our data showed significantly higher cGAMP levels in plasma of SLE patients, compared to those of healthy donors (Fig. 1a). Notably, no such elevation was observed in peripheral blood samples from patients with rheumatoid arthritis (RA) or dermatomyositis (DM) (Fig. 1a). This finding was further confirmed in an independent validation cohort comprising 44 healthy donors and 44 SLE patients (Supplementary Fig. 1a, b). These observations imply that cGAS activation is involved in SLE and may contribute to the pathogenic processes of this disease.

Although SLE is recognized as a complex autoimmune disorder with a multifactorial etiology, a growing number of evidence suggest that TLR7 is a driving factor for human SLE[30-32]. For example, a functional polymorphism of TLR7 was identified predisposing to SLE in humans[30]. The unrestricted TLR7 signaling is associated with human lupus[31]. Moreover, a gain-of-function mutation in TLR7 (Y264H) has been reported to cause human SLE[32]. Therefore, we sought to investigate whether cGAS plays a role in TLR7-mediated SLE. We first analyzed the correlation between *TLR7* expression and cGAMP levels in samples from the above cohort (Supplementary Fig. 1a) and found that the samples with higher *TLR7* expression levels exhibited higher abundance of plasma cGAMP (Fig. 1b). We further found that the TLR7 agonist, R837 or R848[33], stimulated cGAS activation in cells (Fig. 1c and

Supplementary Fig. 1c). These findings indicate the potential interplay between cGAS and TLR7 signaling.

In mice, the exclusive topical application of the TLR7 agonist, imiquimod (IMQ)[33], has been shown to induce lupus-like symptoms[34]. Employing this model, we administered IMQ to wild-type (WT) mice three times a week for a duration of four weeks. We first measured the plasma cGAMP levels in mice and found that IMQ treatment led to a rapid increase in plasma cGAMP levels (Fig. 1d, e), suggesting that cGAS activation is an early event upon TLR7 engagement. During this procedure, we monitored the representative phenotypes of SLE. Our results indicated that while the body weights remained stable (Supplementary Fig. 1d), there was a notable enlargement of the spleen (Supplementary Fig. 1e), and a progressive increase in serum levels of antinuclear antibody (ANA), anti-double-stranded DNA (dsDNA) antibody and IFN-α was observed (Supplementary Fig. 1f–h). The IMQ-induced SLE mice also exhibited renal pathological manifestations, indicated by the IgG and IgM depositions within the glomeruli (Supplementary Fig. 1i) and the pronounced inflammatory cell infiltration (Supplementary Fig. 1j). Strikingly, when *Cgas*[-/-] mice were treated with IMQ, the splenomegaly (Fig. 1f, g and Supplementary Fig. 1k) and other SLE-associated indicators were significantly reduced compared to that of WT mice (Fig. 1h–k). *Cgas* deletion also alleviated the inflammatory cell infiltration induced by IMQ treatment (Fig. 1l). The full kidney histological images from independent mice showed consistent results (Supplementary Fig. 1l). According to a previous report[28], cGAS deletion did not alleviate the autoimmunity in pristane-induced SLE or MRL/*lpr* mice. In our study we obtained similar results (Supplementary Fig. 1a–i). These data indicates that cGAS plays an important role in TLR7-mediated SLE pathogenesis.

We then explored how cGAS is activated in TLR7 signaling. We found that the TLR7-stimulated IFN-α production is markedly dampened in the absence of cGAS in both bone marrow cells and plasmacytoid dendritic cells (pDC) (Fig. 1m, n and Supplementary Fig. 2j–l). We further used ruxolitinib, a Janus Kinase (JAK) inhibitor[35], to block the IFN cascade downstream of TLR7. Under such condition, TLR7 agonist stimuli still effectively triggered the synthesis of cGAMP (Fig. 1o), suggesting that the involvement of cGAS in TLR7 signaling is independent of the feedback of IFN axis. In contrast, we observed that R837/R848 stimuli failed to induce cGAMP production in *Tlr7* deletion cells (Supplementary Fig. 2m, n). We also used RU.521, a cGAS inhibitor[36], and M5049, a TLR7 inhibitor[37], and found that the inhibition of either cGAS or TLR7 attenuated cGAMP production stimulated by R837 or R848 (Supplementary Fig. 2o). We next treated *Tlr7*[-/-] bone marrow cells with RU.521 or treated *Cgas*[-/-] bone marrow cells with M5049 and found that the blockade of both proteins with each strategy dampened the R837/R848-induced cGAMP production (Supplementary Fig. 2p). Together, TLR7 engagement triggers cGAS activation.

Considering that cGAS is a DNA sensor, and mtDNA is known to be released into cytosol under several pathological conditions to activate cGAS[4,17,18], we reasoned that cGAS may also be activated by mtDNA downstream TLR7 engagement. To test this, we quantified the mtDNA abundance in cytosolic fractions and found that TLR7 activation led to a significantly increased level of cytoplasmic mtDNA (Supplementary Fig. 3a, b). A previous study reported that alterations in mitochondrial permeability significantly influence the release of mtDNA[17]. As indicated by the openness of the inner membrane molecular machinery, mitochondrial permeability transition pore (mPTP)[17], R837 or R848 treatment induced an increased mitochondrial permeability (Supplementary Fig. 3c–e).

The BCL2 antagonist/killer 1 (BAK) and BCL2 associated X, apoptosis regulator (BAX) macropores on the outer mitochondrial membrane are key channels for the releasement of mtDNA[38]. Using the BAK/BAX oligomerization inhibitor, MSN-125[39], we found that BAK/BAX

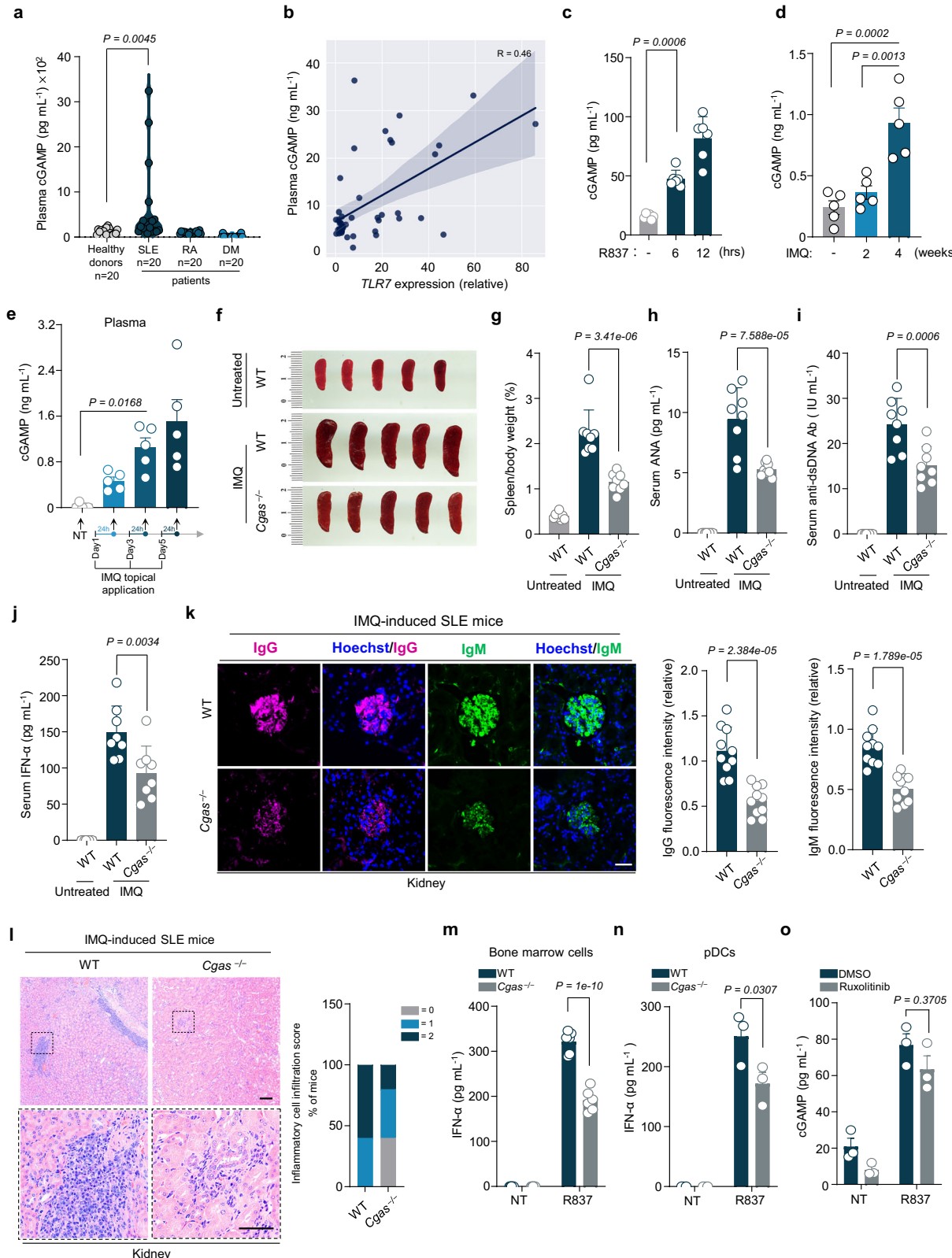

inhibition robustly reduced the mtDNA release (Supplementary Fig. 3f, g) and the IFN-α production (Supplementary Fig. 3h) that are induced by TLR7 activation. Thus, TLR7 activation facilitated the release of mtDNA into the cytosol, thereby triggering cGAS activation and the subsequent IFN-α production. Collectively, these data indicated the important role of cGAS in mediating the IFN production downstream of TLR7.

## Prasugrel blocks cGAS activation by acetylation

Given that our findings imply cGAS inhibition as a potential therapeutic strategy for SLE, we explored approaches to inhibit cGAS. We focused on seeking small molecules capable of inducing cGAS acetylation, since our previous work reported the acetylation-mediated cGAS inhibition[27]. In a compound library comprising 3159 FDA-approved small molecule drugs, we first selected 167 molecules that

**Fig. 1 | cGAS mediates the autoimmunity in TLR7-driven systemic lupus erythematosus. a** Plasma cGAMP concentrations of healthy donors ($n = 20$), SLE patients ($n = 20$), RA patients ($n = 20$) and DM patients ($n = 20$). **b** Correlation analysis between plasma cGAMP levels and *TLR7* expression of SLE patients ($n = 44$). **c** Analysis of cGAMP in bone marrow cells at the indicated times after R837 treatment (n = 6 independent biological replicates). **d, e** Plasma cGAMP levels of IMQ-induced mice at indicated time points ($n = 5$ independent biological replicates). Representative images of spleens of WT and *Cgas*$^{-/-}$ mice treated with IMQ for 4 weeks (**f**), and the ratio of spleen to body weight was calculated (**g**) ($n = 8$ independent biological replicates). ELISA detection of serum anti-nuclear antibodies (**h**), anti-dsDNA antibodies (**i**) and IFN-α concentrations (**j**) of WT and *Cgas*$^{-/-}$ BALB/c mice treated with IMQ for 4 weeks ($n = 8$ independent biological replicates). **k** Immunofluorescent staining of IgG (purple) and IgM (green) in kidneys from WT and *Cgas*$^{-/-}$ mice treated with IMQ for 4 weeks. Representative images are shown.

Scale bar, 20 μm. The quantitative analysis was performed ($n = 10$ independent biological replicates). **l** HE-stained section of kidney from WT and *Cgas*$^{-/-}$ mice treated with IMQ for 4 weeks. Scale bar, 200 μm (top) and 50 μm (bottom). Representative images are shown. Percentage of the quantitative inflammatory cell infiltration scores ($n = 5$ per group). Bone marrow cells (**m**, 6 independent biological replicates) or pDCs (**n**, 3 independent biological replicates) from WT and *Cgas*$^{-/-}$ mice were treated with R837 and the IFN-α concentrations were measured. **o** Analysis of cGAMP in bone marrow cells pretreated as indicated and then treated with R837 ($n = 3$ independent biological replicates). NT, non-treated (**e, m, n, o**). Statistical significance was determined using one-way ANOVA with Tukey's test (**a, c–e, g–j**); two-tailed student's unpaired *t*-test (**k**); two-way ANOVA with Tukey's test (**m–o**). Data are presented as the means ± SEM (**c–e, g–k, m–o**). *P* values are shown with figures. Source data are provided as a Source Data file.

contained an acetyl group, and then performed in silico docking simulations with cGAS (PDB: 4O67) using Discovery Studio 2022 (Fig. 2a). Among them, 106 molecules with potential cGAS-binding capabilities were ranked according to their CDOCKER interaction energy (Supplementary Table 1). The top 15 candidates from this screening were then experimentally verified for their ability to acetylate cGAS. Using aspirin as a positive control[27], we found that prasugrel was the only agent that induced cGAS acetylation, specifically at the key regulatory sites, Lys384, Lys394 and Lys414, which are known to inactivate cGAS upon acetylation (Fig. 2b). The docking model showed the special accessibility of prasugrel to the key regulatory lysine sites of cGAS (Fig. 2c–f).

We further examined the prasugrel-mediated cGAS acetylation and found that compared to aspirin, prasugrel can achieve a similar cGAS acetylation effect at one-thousandth of the concentration of aspirin (Fig. 2g and Supplementary Fig. 4a). With a human monocytic cell line, U937, we demonstrated that prasugrel (100 μM) robustly acetylated endogenous cGAS in cells (Fig. 2h). As expected, prasugrel treatment significantly suppressed the DNA-stimulated cGAS activation (Fig. 2i), IRF3 phosphorylation (Fig. 2j) and IFN production (Fig. 2k, l). At this concentration, prasugrel had no cytotoxicity effect (Supplementary Fig. 4b), and did not inhibit cGAMP-triggered downstream events (Fig.2m and Supplementary Fig. 4c). These data suggested that prasugrel specifically inhibits the DNA-induced IFN production by acetylating cGAS. We then challenged U937 cells with intracellular RNA mimics, poly(I:C), a synthetic analogue of dsRNA that activates intracellular RNA-sensing receptors[40–42]. Our data indicated that prasugrel did not inhibit poly(I:C)-induced IFN production signaling (Fig. 2n and Supplementary Fig. 4d, e). These findings were further confirmed in murine bone marrow-derived macrophages (BMDM) (Fig. 2o, p and Supplementary Fig. 4f). We also treated BMDMs with TLR1/2 agonist, Pam3CSK4, and TLR4 agonist, LPS. Our data showed that prasugrel did not suppress the expression of *Il6* downstream of these TLRs (Supplementary Fig. 4g, h). Additionally, we examined the effect of prasugrel in cGAS-deficient cells and found that prasugrel had no inhibitory effect on the residual *IFNB1* expression stimulated by DNA (Supplementary Fig. 4i). These data collectively suggest that prasugrel inhibits DNA-induced cGAS activation with certain specificity.

**Acetylation disrupts DNA-induced phase condensation of cGAS**
Recent studies highlighted phase condensation as a crucial step in cGAS activation[43,44]. To understand how acetylation inhibits cGAS activation, we examined the effect of acetylation on cGAS phase condensation. By incubating the purified recombinant protein cGAS$^{Non-Ac}$ and the site-specific acetylated variants, cGAS$^{K384Ac}$, cGAS$^{K394Ac}$ and cGAS$^{K414Ac}$ (Fig. 3a) with FAM-labeled double-stranded DNA (dsDNA), we observed a robust and rapid condensate formation of DNA and cGAS$^{Non-Ac}$. In contrast, the acetylated cGAS exhibited a significant reduction in liquid droplet formation (Fig. 3b–e and Supplementary

Fig. 5a). Because we previously reported that acetylation of cGAS at single-site (K384, K394 and K414) had only a limited effect on DNA-binding[27], we next used PEG-8000 to promote condensate formation of cGAS proteins. We found that the unmodified cGAS proteins formed more robust condensates than those of acetylated variants (Supplementary Fig. 5b), indicating that acetylation attenuates the phase separation capacity of cGAS. We further assessed whether prasugrel could inhibit the DNA-triggered phase condensation of cGAS. Recombinant cGAS$^{Non-Ac}$ proteins were pretreated with prasugrel, G140, an inhibitor that occupies the ATP/GTP-binding site of cGAS[45] or XQ2B, an inhibitor targeting cGAS–DNA binding[46], before the incubation with DNA. Our data showed that while all three chemicals strongly suppressed cGAS activation (Fig. 3f), only prasugrel and XQ2B restrained the formation of the cGAS condensates (Fig. 3g, h). Consistently, using cell lines stably expressing mEGFP-tagged cGAS (cGAS-mEGFP), we show that prasugrel or XQ2B significantly reduced the formation of DNA-stimulated cGAS droplets in cells (Fig. 3i, j and Supplementary Fig. 5c). These results demonstrate that acetylation disturbs the DNA-induced phase condensation of cGAS, thereby blocking cGAS activation.

**Prasugrel suppresses cGAS-dependent autoimmune responses**
We next examined whether prasugrel could be used to treat cGAS-mediated autoimmune responses. To do so, we used the *Trex1*-deficient model. We first deleted *Trex1* in L929 cells to induce the expression of interferon-stimulated genes (ISG), indicators of autoimmune responses in cells[16,19]. Prasugrel effectively inhibited the expression of ISGs (Fig. 4a, b). We further evaluated the effect of prasugrel using primary bone marrow cells from *Trex1*-deficient mice and obtained consistent results (Fig. 4c, d). The half-maximal inhibitory concentration (IC$_{50}$) of prasugrel for inhibiting ISG expression was calculated (Fig. 4d). A daily administration of prasugrel (100 mg/kg) for 14 days showed that this compound was well tolerated by mice (Fig. 4e). According to previous publications[16,19,27], we used the ISG expression levels in the heart tissues of the *Trex1*$^{-/-}$ mice as the indicator for disease severity[16,19,27], we showed that the daily intramuscular injection of prasugrel at 20 mg/kg for 7 days significantly reduced ISG expression in the hearts (Fig. 4f). Furthermore, prasugrel treatment prolonged the survival of *Trex1*$^{-/-}$ mice (Fig. 4g). Thus, prasugrel exhibits promising effect in alleviating the autoimmunity in mice by inhibiting cGAS.

**Prasugrel ameliorates autoimmunity in SLE mice**
We then examined the effectiveness of prasugrel using the IMQ-induced mouse model. We gave daily intramuscular injections of prasugrel to IMQ-induced mice for 4 weeks and collected the sera and tissues for analysis (Fig. 5a). To rule out the possibility that the inhibitory effect of prasugrel could also be mediated through its known target, the P2Y12 receptor[47], we utilized clopidogrel, another P2Y12-targeting medication used for treating clotting[47], as

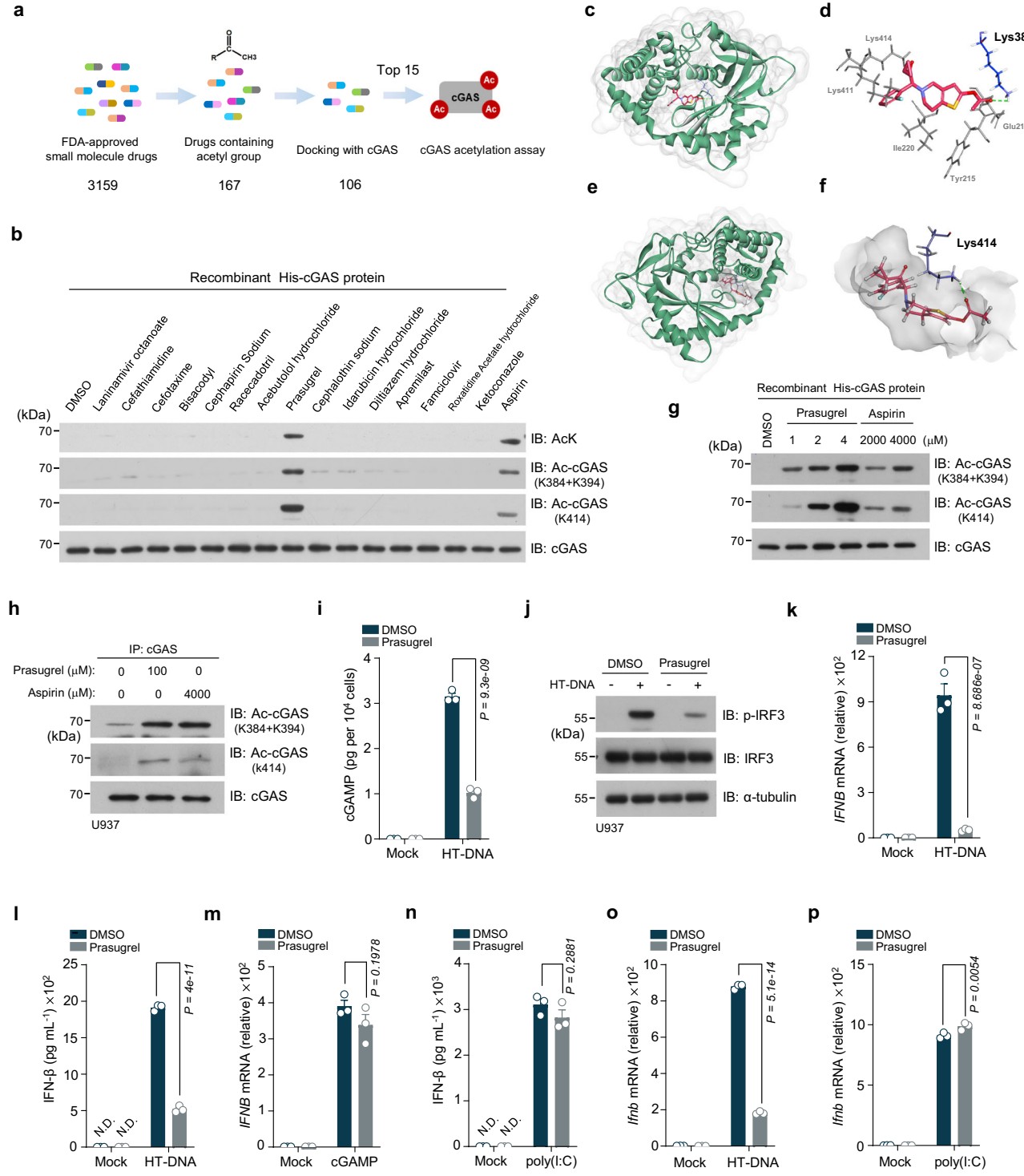

comparative control. Our data showed that prasugrel significantly reduced the splenomegaly (Fig. 5b, c), serum levels of autoantibodies (Fig. 5d, e) and serum IFN-α levels (Fig. 5f), compared to that of clopidogrel treated group. In addition, the renal pathological manifestations were also relieved by prasugrel (Fig. 5g–i). This trend was also confirmed by full kidney histological images from independent mice (Supplementary Fig. 6a). Because cGAS deletion did not alleviate the autoimmunity in MRL/*lpr* mice (Supplementary Fig. 2a–d), prasugrel treatment showed consistent results in these mice (Supplementary Fig. 6b–d). Collectively, these data suggest that prasugrel ameliorates the autoimmunity in SLE mice, likely independent of its known target, P2Y12.

## Prasugrel exhibits potential therapeutic effects in SLE patient cells

To further demonstrate the potential therapeutic effect of prasugrel, we obtained peripheral blood mononuclear cells (PBMC) from a cohort with 10 SLE patients (Fig. 6a). A concentration of 50 μM of prasugrel, which was determined as a non-toxic dosage (Fig. 6b), was used to treat PBMCs before an RNA-seq analysis. Because the systemic disease characteristics of SLE were unable to evaluate in patient cells, we used several published IFN signatures to reflect the SLE-associated immune responses in these cells[32,48–50]. Significantly, prasugrel ameliorated the SLE-associated immune responses in patient cells, as indicated by the expression change of IFN signatures. As a positive

**Fig. 2 | Prasugrel inhibits cGAS activation through acetylation. a** A diagram of the process of screening for small molecule drugs with acetylation potential. **b** Immunoblot analysis of cGAS acetylation by indicated drugs with the pan-acetyl-lysine antibody (AcK) and site-specific cGAS acetylation antibodies (Ac-cGAS: K384 + K394; Ac-cGAS: K414), cGAS protein, 300 nM per group. Representative of two independent experiments. Structural views of prasugrel docking with cGAS (Lys384) (**c, d**) and cGAS (Lys414) (**e, f**). **g** Immunoblot analysis of cGAS acetylation incubated with prasugrel and aspirin using site-specific cGAS acetylation antibodies, cGAS protein, 300 nM per group. Representative of three independent experiments. **h** Immunoprecipitation (IP) analysis of cGAS acetylation in PMA-differentiated U937 cells treated with DMSO, prasugrel or aspirin for 24 h. Representative of three independent experiments. **i** U937 cells (PMA-differentiated, hereinafter the same unless otherwise indicated) were pretreated with DMSO or prasugrel before transfected with herring testis (HT-DNA). cGAMP concentrations

were analyzed by LC-MS/MS. Data are presented as the means ± SEM from three independent experiments. **j** Immunoblot analysis of indicated proteins in U937 cells transfected with HT-DNA following the pretreatment with DMSO or prasugrel for 24 h. Representative of three independent experiments. U937 cells were pretreated with DMSO or prasugrel for 24 h, followed by the HT-DNA transfection (**k, l**), cGAMP treatment (**m**) or poly(I:C) transfection (**n**). qPCR analysis of *IFNB* mRNA expression (**k, m**) or ELISA analysis of IFN-β secretion (**l, n**) as indicated. Data are presented as the means ± SEM from three independent experiments. qPCR analysis of *Ifnb* mRNA expression in BMDM transfected with HT-DNA (**o**) or poly(I:C) (**p**) following the pretreatment with DMSO or prasugrel for 24 h. Data are presented as the means ± SEM from three independent experiments. N.D. not detected (**l, n**). Statistical significance was determined using two-way ANOVA with Tukey's test (**i, k–p**). *P* values are shown with figures. Source data are provided as a Source Data file.

control, the JAK1/2 inhibitor ruxolitinib showed a comparable effect (Fig. 6c).

To determine whether cGAS activation correlates with the inhibitory effect of prasugrel, we further collected PBMCs and plasma from an additional cohort of 26 SLE patients (Fig. 6d). In addition to examine the IFN signature change in patient PBMCs that treated with or without prasugrel, we also measured the plasma cGAMP levels. Importantly, our analysis revealed that the expression of IFN signature genes was substantially more suppressed in SLE patients exhibiting elevated plasma cGAMP levels (Fig. 6e, f). This result was further validated using three additional IFN signatures (Supplementary Fig. 7a–f), according to previous reports[32,49,50]. These findings suggest that plasma cGAMP levels could serve as a predictive biomarker to identify SLE patients who are likely to respond favorably to prasugrel treatment. Collectively, our work presents prasugrel as a therapeutic candidate for SLE. Given its status as an FDA-approved medication, our findings also underscore its immediate clinical translation.

## Discussion

Accumulating evidence indicates the significant association between the aberrant activation of cGAS and SLE[4,5,16,19,29]. For instance, approximately 1–2% of SLE patients harbor *TREX1* mutations[23], which results in the accumulation of self-DNA and the subsequent IFN production through the activation of cGAS[16,19]. Similarly, a recent investigation revealed increased cGAMP levels in PBMCs in 7 of 48 SLE patients, suggesting the activation of cGAS in these individuals[29]. Despite these findings, the precise role and contribution of cGAS in human SLE still remain obscure. In this study, we show that the plasma cGAMP level is elevated in patients with SLE, but not RA or DM, indicating a specific role of cGAS in SLE. Notably, samples with elevated *TLR7* expression levels correlate with increased plasma cGAMP concentrations. Utilizing a TLR7-driven SLE mouse model, we demonstrate that the deletion of cGAS mitigates lupus-like symptoms. Thus, our findings uncover a novel interplay between TLR7 signaling and cGAS activation in SLE pathogenesis.

Current standard therapies for SLE, including glucocorticoids, hydroxychloroquine, and immunosuppressive agents, broadly impact the immune systems. These treatments lack precise targets, leading to various side effects and limited efficacy[3]. There is a pressing need to deepen our understanding of SLE pathogenesis to identify reliable biomarkers and develop novel therapeutic strategies. The selective JAK inhibitors are considered targeted therapies for SLE and are currently in clinical trials[51]. Directly targeting cGAS could be an alternative and potentially more effective approach to treat SLE, as cGAS mediates the IFN production, an upstream event for JAK activation. Although plasma cGAMP was detected in only a subset of patients, the assessment was performed at a single time point. Given the complexity of SLE as a systemic disease, the activation of cGAS may vary across different stages of the disease. Therefore, it is possible that cGAS activation is relevant to a larger patient population. Thus, therapeutic strategies

targeting cGAS have the potential to benefit a broader range of SLE patients.

We previously reported that the acetylation of cGAS at either Lys384, Lys394 or Lys414 blocks its activation, offering a viable strategy for cGAS targeting[27]. In this study, we further find that acetylation disrupts DNA-triggered liquid phase condensation of cGAS, a process essential for cGAS activation[43,44]. By inhibiting condensate formation, acetylation on cGAS efficiently attenuates the downstream type I IFN production. This finding indicates that the lysine acetylation is likely to act as a molecular switch that controls cGAS activity. A notable discovery of our study is the identification of prasugrel as a potent cGAS inhibitor. This inhibition is achieved by acetylation. While post-translational modifications such as phosphorylation are commonly used as drug targets for therapeutic interventions[52], the acetylation-mediated mechanism remains to be further explored. Our study thus pioneers a novel paradigm of drug discovery based on protein acetylation.

Notably, the treatment with prasugrel results in a significantly more reduction in the expression of IFN signature genes in PBMCs among patients exhibiting higher plasma cGAMP levels. This correlation suggests that elevated plasma cGAMP could serve as a predictive biomarker for identifying SLE patients who are more likely to respond to prasugrel treatment. Importantly, a significant proportion of SLE patients also develop antiphospholipid syndrome (APS), an autoimmune disorder that increases the risk of blood clot formation[53]. Prasugrel may offer dual therapeutic benefits for SLE patients with coexisting APS, because it is a commonly used anti-platelet medication for blood clot prevention[47]. Considering that prasugrel is an FDA-approved medication with an established safety profile, its repurposing for SLE treatment represents a worthwhile avenue to explore.

An increasing number of studies indicate that cGAS drives inflammation and acts as a key factor promoting aging and various diseases[7,15,16,19,38,54–57]. Our study may contribute to the management of conditions and diseases related to cGAS activation beyond SLE. Our work also highlights the importance of further clinical trials for prasugrel in treating SLE and other cGAS-related diseases.

## Methods
### Mice
*Trex1*[+/−] of C57BL/6 background were from D. Barnes and T. Lindahl (Cancer Research UK). Wild-type and *Trex1*[−/−] mice (3-week-old) were given daily (i.m.) injection of prasugrel (20 mg/kg). *Cgas*[+/−] mice were generated in our laboratory. Knockout of the *Cgas* gene in mice was achieved using CRISPR-Cas9 technology. The following sgRNA sequences were used: sgRNA1: 5′-GTGAGGTCTTGCCAGTAGAG−3′, sgRNA2: 5′- TGCTCTTCGGAGAGTAGGCC−3′). MRL (000486) and MRL/*lpr* (000485) mice were purchased from The Jackson Laboratory. *Cgas*[+/−] MRL/*lpr* were generated from Cyagen Biosciences (sgRNA1: 5′-TTCGAAGAAAGGCCGCGAAA−3′, sgRNA2: 5′-GAAGTGTGTGTCACCG

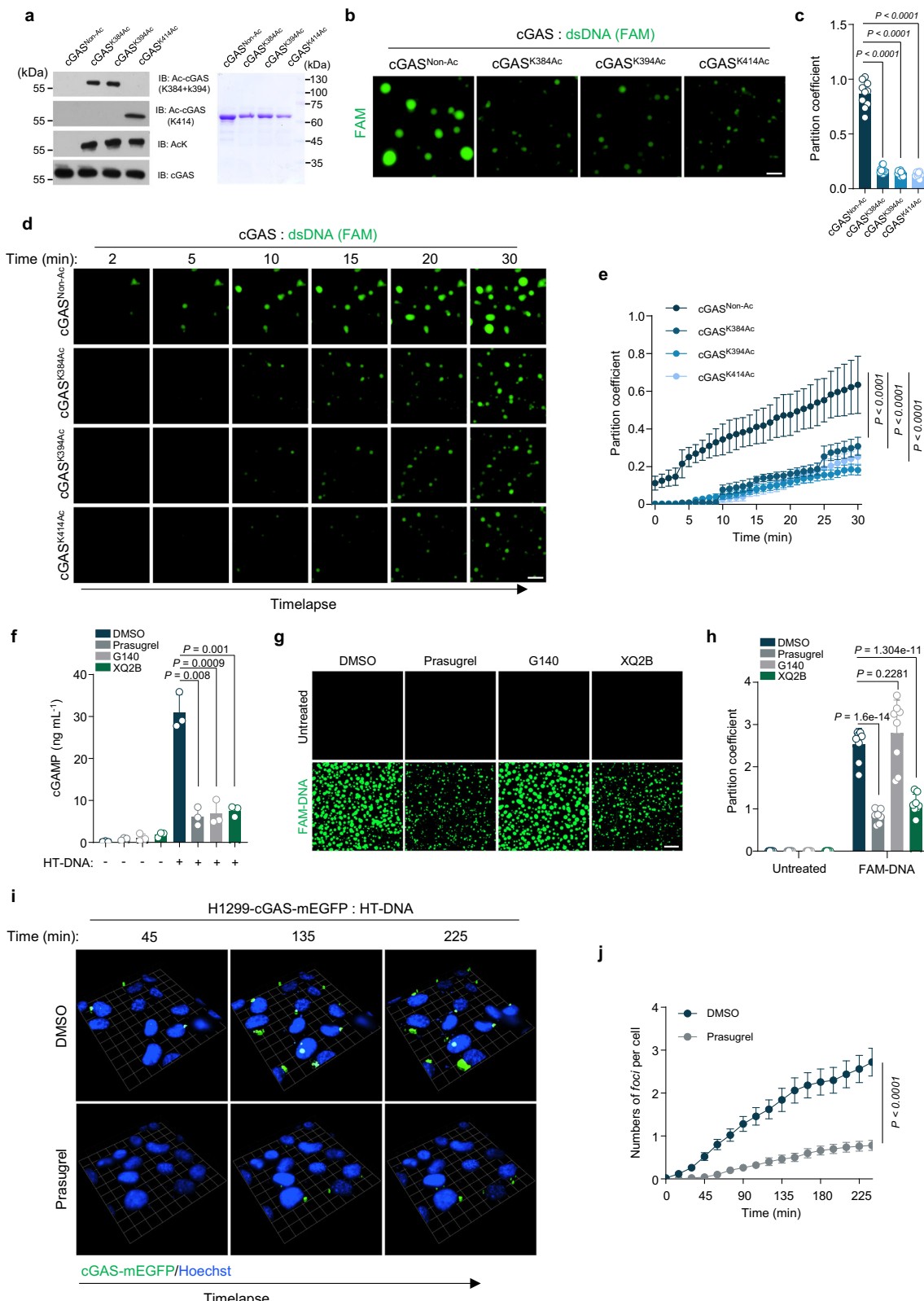

CCATAG−3′, sgRNA3: 5′-ATATATGGCGGGAACGTAGC−3′, sgRNA4: 5′-AAGCAGGTTTCGTGTACCTC−3′). 8–16-week-old females of MRL/MpJ and MRL/*lpr* were used in the study. *Tlr7*[+/−] mice (S-KO-02979) were purchased from Cyagen Biosciences. All mice were maintained under controlled temperature (22 ± 1 °C) and humidity (50–60%) with a 12 h light-dark cycle, and housed in a non-specific pathogen-free facility with ad libitum access to food and water. Throughout the study, experimental and control animals were bred and housed separately. Euthanasia was performed by carbon dioxide inhalation. All animal experiments were performed in accordance with the National Institutes of Health Guide for the Care and Use of Laboratory Animals and with the approval of the Institutional Animal Care and Use Committee (No. IACUC-DWZX-2022-720).

**Fig. 3 | Acetylation disrupts DNA-induced liquid phase condensation of cGAS.
a** Immunoblot analysis of the acetylated recombinant cGAS proteins with site-
specific cGAS acetylation antibodies and the anti-pan-acetyl-lysine antibody (left).
Coomassie brilliant blue-stained SDS-PAGE gel of purified WT and different site-
specific acetylated cGAS proteins (right). Representative of three independent
experiments. **b**−**e** FAM-labeled dsDNA were incubated with recombinant cGAS$^{Non-Ac}$,
cGAS$^{K384Ac}$, cGAS$^{K394Ac}$ and cGAS$^{K414Ac}$ proteins, 10 µM per group. The liquid phase
condensations of cGAS proteins were observed with fluorescence microscope (**b**)
or Time-lapse imaging (**d**). Representative of three independent experiments. Scale
bar, 10 µm. The quantitative analysis was performed (**c**, **e**). Data are presented as
the means ± SD from three independent experiments. **f**−**h** Recombinant cGAS
proteins (10 µM) were incubated with DMSO, prasugrel, G140 or XQ2B, followed by
DNA treatment. The DNA-stimulated cGAMP synthesis in vitro were measured by
ELISA (**f**). Data are presented as the means ± SEM from three independent experi-
ments. The liquid phase condensations of cGAS proteins were observed with
fluorescence microscope (**g**) and quantitatively analyzed (**h**). Representative of
three independent experiments. Scale bar, 10 µm. Data are presented as the
means ± SEM from three independent experiments. **i**, **j** Time-lapse images (**i**) and
quantitative analysis (**j**) of cGAS-mEGFP *foci* in H1299-cGAS-mEGFP cells trans-
fected with HT-DNA in the presence of DMSO or prasugrel, 1 unit = 7.59 µm.
Representative of three independent experiments. Data are presented as the
means ± SEM from three independent experiments. Statistical significance was
determined using one-way ANOVA with Tukey's test (**c**); two-way ANOVA with
Tukey's test (**e**, **f**, **h**, **j**). *P* values are shown with figures. Source data are provided as a
Source Data file.

## Imiquimod-induced lupus model
All BALB/c mice used in the imiquimod-induced lupus model were 8-
week-old females. The skin on the ears of the mice was treated with 5%
imiquimod cream, 3 times weekly. Mice serum was collected and
measured for autoantibodies and IFN-α. Kidneys were used to evaluate
the deposition of IgG and IgM for tissue immunofluorescence and
inflammatory infiltration for HE staining.

## Pristane-induced lupus model
12–16-week-old females C57BL/6 N mice received a single intraper-
itoneal injection of 0.5 mL of pristane and serum samples were col-
lected for analyzing autoantibodies and IFN-α level 6 months after
pristane injection.

## Cell culture
U937 (CRL-1593.2, ATCC), L929 (CCL-1, ATCC), H1299 (CRL-5803,
ATCC) and pDCs were cultured in RPMI 1640 (CM10040, MACGENE)
supplemented with 10% Fetal Bovine Serum (FBS, FSD500, EXCELL), 1%
penicillin-streptomycin (CC004, MACGENE). BMDMs and bone mar-
row cells were cultured in DMEM (CM10013, MACGENE) supplemented
with 10% FBS, 2 mM glutamine (CC009, MACGENE), 1% streptomycin-
penicillin. Cells were cultured in a 5% $CO_2$ incubator at 37 °C. U937 cells
were differentiated with PMA (0.1 µM) (524400, Sigma-Aldrich) for
36–48 h before transfection or other treatment. pDCs were isolated
from spleen of mice using plasmacytoid dendritic cell isolation kit
(130-107-093, Miltenyi Biotec). Bone marrow cells were isolated from
the femurs of mice and RBCs were removed using RBC Lysis Buffer
(A1049201, Thermo Fisher Scientific) before stimulating. BMDMs were
differentiated from bone marrow cells with recombinant mouse M-CSF
(25 ng/mL, 416-ML-050, R&D systems) for 7 days. All cell lines were
tested to be mycoplasma free by PCR.

Human PBMCs were isolated from whole blood of consenting
donors with Histopaque-1077 (10771, Sigma-Aldrich) according to the
manufacturer's instructions. Informed consent was obtained from the
blood donors by the institutions. The study was approved by the ethics
committee of Jiaxing Central Blood Station (No. 2024-005), the ethics
committee of Peking University People's Hospital (No. 2019PHB089-
01) and the Renji Hospital Ethics Committee of Shanghai Jiao tong
University School of Medicine (No.2013-126). It was conducted in
compliance with all relevant ethical regulations.

## Antibodies
For western blotting, antibodies against acetylated-Lysine (9441 s,
RRID: AB_331805), cGAS (mouse, 31659 s, RRID: AB_2799008) and
TLR7 (82658 s, RRID: AB_3662102) were obtained from Cell Signaling
Technology. Antibodies against IRF3 (ab68481, RRID: AB_11155653) and
p-IRF3 (ab76493, RRID: AB_1523836) were from Abcam. Antibodies
against cGAS, Ac-cGAS (K384 + K394) and Ac-cGAS (K414) were gen-
erated from our laboratory and validated in our previous work[27].
Antibodies against β-actin (66009-1-Ig, RRID: AB_2687938), GAPDH
(10494-1-AP, RRID: AB_2263076) and α-Tubulin (66031-1-Ig, RRID:

AB_11042766) were purchased from Proteintech Groups. Antibody
against TREX1 (611986, RRID: AB_399407) was from BD Biosciences.
For Immunofluorescence, Goat Anti-Mouse IgM mu chain (Alexa Fluor
488) (ab150121, RRID: AB_2801490) was from Abcam. Goat anti-Mouse
IgG (H + L) Highly Cross-Adsorbed Secondary Antibody (Alexa Fluor
647) (A21236, RRID: AB_2535805) was from Invitrogen. Antibody
against G3BP1 (13057-2-AP, RRID: AB_2232034) was from Proteintech
Groups. For fluorescence-activated cell sorting (FACS) analysis, PE anti-
mouse CD11c antibody (117307, RRID: AB_313776) and APC anti-mouse
PDCA-1 antibody (127016, RRID: AB_1967127) were from Biolegend. The
antibody dilution rates are provided in Supplementary Table 2.

## Reagents
Laninamivir octanoate (T9124), cefathiamidine (T8134), cefotaxime
(T7733), bisacodyl (T6414), cephapirin sodium (T5038), racecadotril
(T1176), acebutolol hydrochloride (T1012), cephalothin sodium
(T1122), idarubicin hydrochloride (T6010), diltiazem hydrochloride
(T0112), apremilast (T2923), famciclovir (T1646) and roxatidine acet-
ate hydrochloride (T0157) were purchased from TargetMOl Chemicals
Inc. Prasugrel (HY-15284), clopidogrel (HY-15283), R837 (HY-B0180),
R848 (HY-13740), PEG-8000 (HY-Y0873J), G140 (HY-133916), XQ2B
(HY-P5997), ruxolitinib (HY-50856), MSN-125 (HY-120079) and PEG-
8000 (HY-Y0873J) were from MedChemExpress. Aspirin (A2093),
pristane (P2870) and HT-DNA (D6898) were purchased from Sigma-
Aldrich. RU.521 (S6841) and M5049 (S9931) were from Selleck. Imi-
quimod was from Mingxin Pharmaceutical Tech. Poly(I:C) (tlrl-pic) and
cGAMP (tlrl-nacga23) were from InvivoGen.

## Cell viability assay
U937, BMDMs or PBMCs were seeded into 96-well plates and incu-
bated with prasugrel at indicated concentrations for 24 h. Cell viability
was analyzed by CellTiter One Solution Cell Proliferation Assay (MTS)
(G3580, Promega) according to the manufacturer's instruction.

## Immunoblotting and immunoprecipitation
PMA-differentiated U937 cells treated with prasugrel or aspirin were
lysed in M2 buffer (20 mM Tris-HCl pH 7.5, 0.5% Nonidet P-40, 250 mM
NaCl, 3 mM EDTA, 3 mM EGTA, 2 mM dithiothreitol) supplemented
with complete protease inhibitor cocktail (HY-K0010, MedChemEx-
press), 1 µM trichostatin A (TSA, HY-15144, MedChemExpress) and
10 mM nicotinamide (NAM, N0636, Sigma-Aldrich), followed by soni-
cation and centrifugation at 15,000 *g* for 10 min at 4 °C. The super-
natants were immunoprecipitated with cGAS antibody for 8 h at 4 °C.
The immunoprecipitants were washed six times with M2 buffer and
boiled in SDS-loading buffer for immunoblot analysis to detect cGAS
acetylation.

## RNA extraction and quantitative PCR
For siRNA-mediated knockdown in L929 cells, siRNAs were transfected
with Lipofectamine RNAiMAX (13778150, Invitrogen) at a final con-
centration of 50 nM. After 24 h siRNA transfection, cells were treated

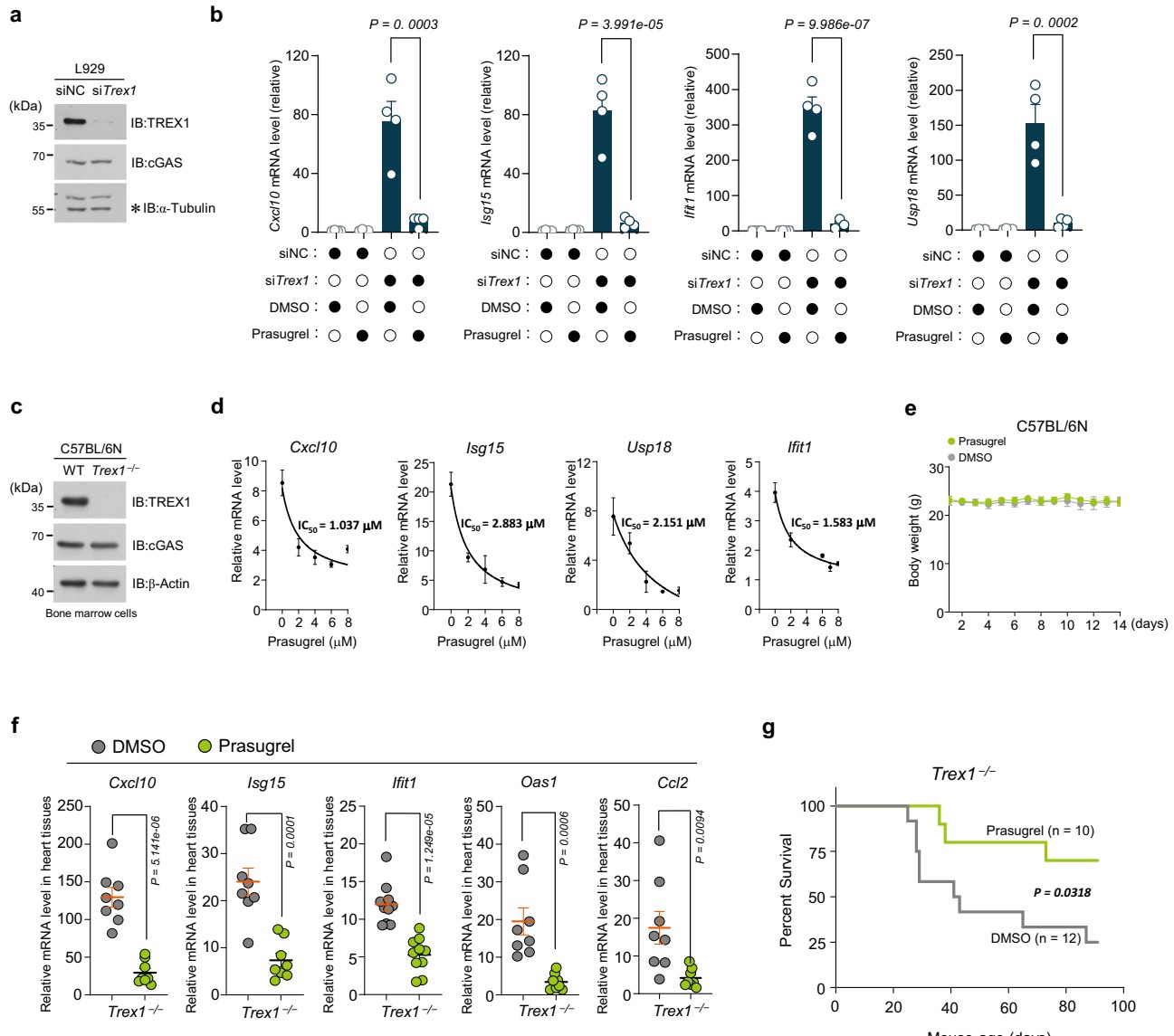

**Fig. 4 | Prasugrel suppresses cGAS-dependent autoimmune responses.** Immunoblot analysis of knockdown effect of in L929 cells with indicated antibodies (**a**) and qPCR analysis of ISGs mRNA expression followed by pretreatment with DMSO or prasugrel for 24 h (**b**). Data are presented as the means ± SEM from four independent experiments. Immunoblot analysis of WT and *Trex1⁻/⁻* bone marrow cells from C57BL/6 N mice with indicated antibodies (**c**) and the mRNA levels of ISGs were analyzed by qPCR as indicated (relative to that of WT bone marrow cells) (**d**). The IC₅₀ was calculated by prism.10.1. Data are presented as the means ± SD from three independent experiments. **e** C57BL/6 N WT mice were given daily administration (i.m.) of DMSO or prasugrel (100 mg/kg) for 14 days. Body weights of mice

were monitored (*n* = 5 biological replicates). Data are presented as the means ± SD. **f** *Trex1⁻/⁻* mice were given daily administration (i.m.) of DMSO or prasugrel (20 mg/kg) for 7 days and then ISGs mRNA in the mouse heart tissues was analyzed by qPCR. Data are presented as the means ± SEM (*n* = 8 independent biological replicates). **g** Survival analysis of 3-weeks old *Trex1⁻/⁻* mice administrated (i.m.) with DMSO or prasugrel daily. NC, negative control (**a**, **b**). Statistical significance was determined using two-way ANOVA with Tukey's test (**b**); two-tailed student's unpaired *t*-test (**f**); log rank (Mantel-Cox) test (**g**). *P* values are shown with figures. Source data are provided as a Source Data file.

with prasugrel for 24 h. Mouse *Trex1* (MSS238570, 5′-ACCGACAGA-CUCACAUACUGCUGAA-3′) siRNA was from Invitrogen. Total RNAs were isolated from cells or tissues using TRI reagent. Total RNAs (500 ng) were reversed-transcribed to cDNA using the PrimeScript RT Reagent Kit (RR037A, Takara). mRNA expression was analyzed by qPCR with PowerUp SYBR Green Master Mix (A25742, Thermo Fisher Scientific) on Applied Biosystems Step OnePlus system. Primers are listed in Supplementary table 3.

### Quantification of cGAMP

Serum of clinical samples or murine were mixed with 800 μL of cold extraction solvent (40:40:20 (v/v/v) methanol-acetonitrile-water) and

then stored at −30 °C for 30 min and centrifuged at 15,000 *g* for 15 min. The supernatants were evaporated at room temperature. For LC-MS/MS quantitation, the pellets were resuspended in 200 μL of ammonium acetate buffer (0.05% acetate in water) and then were submitted to a triple-quadrupole mass spectrometer (Xevo TQ-S, Waters Corp.) equipped with an electrospray ionization source. The nebulizer gas was 99.95% nitrogen, and the collision gas was 99.99% argon with a pressure of $3 \times 10^{-3}$ mbar in the T-Wave cell. The gas flows of the cone and desolvation were set as 150 and 800 l/h, respectively. The cGAMP measurements were performed in the positive mode with a 3.5 kV capillary voltage, 120 °C source temperature and 450 °C desolvation temperature. The optimized ion transitions were: cGAMP m/z

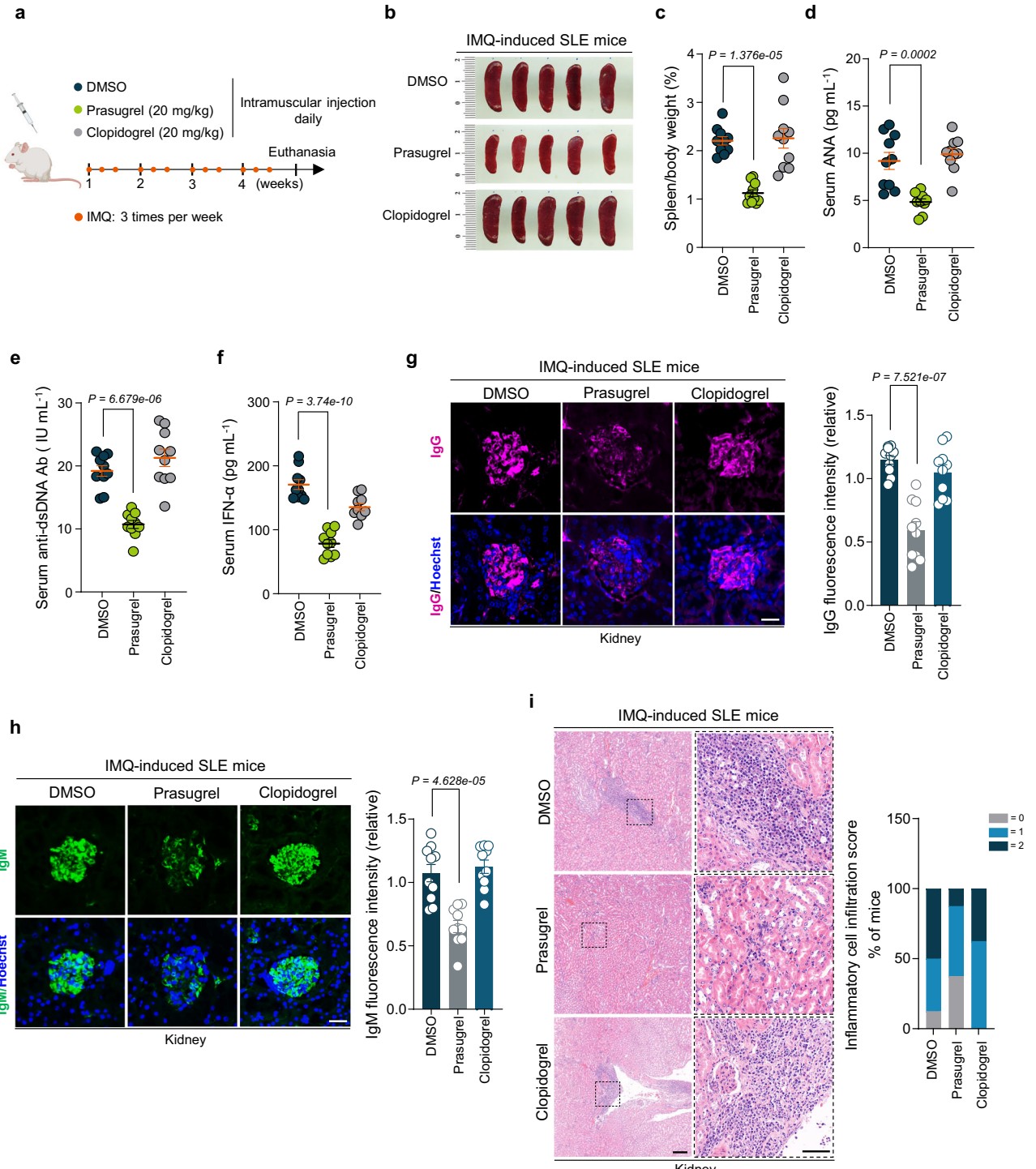

**Fig. 5 | Prasugrel ameliorates autoimmunity in SLE mice. a** Schematic drawing of experimental design and treatment timeline for IMQ-induced SLE model by Med-Peer. BALB/c female mice were treated with prasugrel or clopidogrel following IMQ administration. Representative images of spleens of mice (**b**) and the ratio of spleen to body weight (**c**). Data are presented as the means ± SEM (*n* = 10 independent biological replicates). ELISA detection of serum anti-nuclear antibodies (**d**), anti-dsDNA antibodies (**e**) and IFN-α concentrations (**f**). Data are presented as the means ± SEM (*n* = 10 independent biological replicates). Immunofluorescent staining and quantitative analysis of IgG (purple) (**g**) and IgM (green) (**h**) in kidneys from mice in each group. Representative images are shown. Scale bar, 20 μm. The quantitative analysis was performed. Data are presented as the means ± SEM (*n* = 10 independent biological replicates). **i** HE-stained section of kidneys from the mice in each group. Scale bar, 200 μm (left) and 50 μm (right). Representative images are shown. Percentage of quantitative inflammatory cell infiltration score (*n* = 8 biological replicates). Statistical significance was determined using one-way ANOVA with Tukey's test (**c**–**h**). *P* values are shown with figures. Source data are provided as a Source Data file.

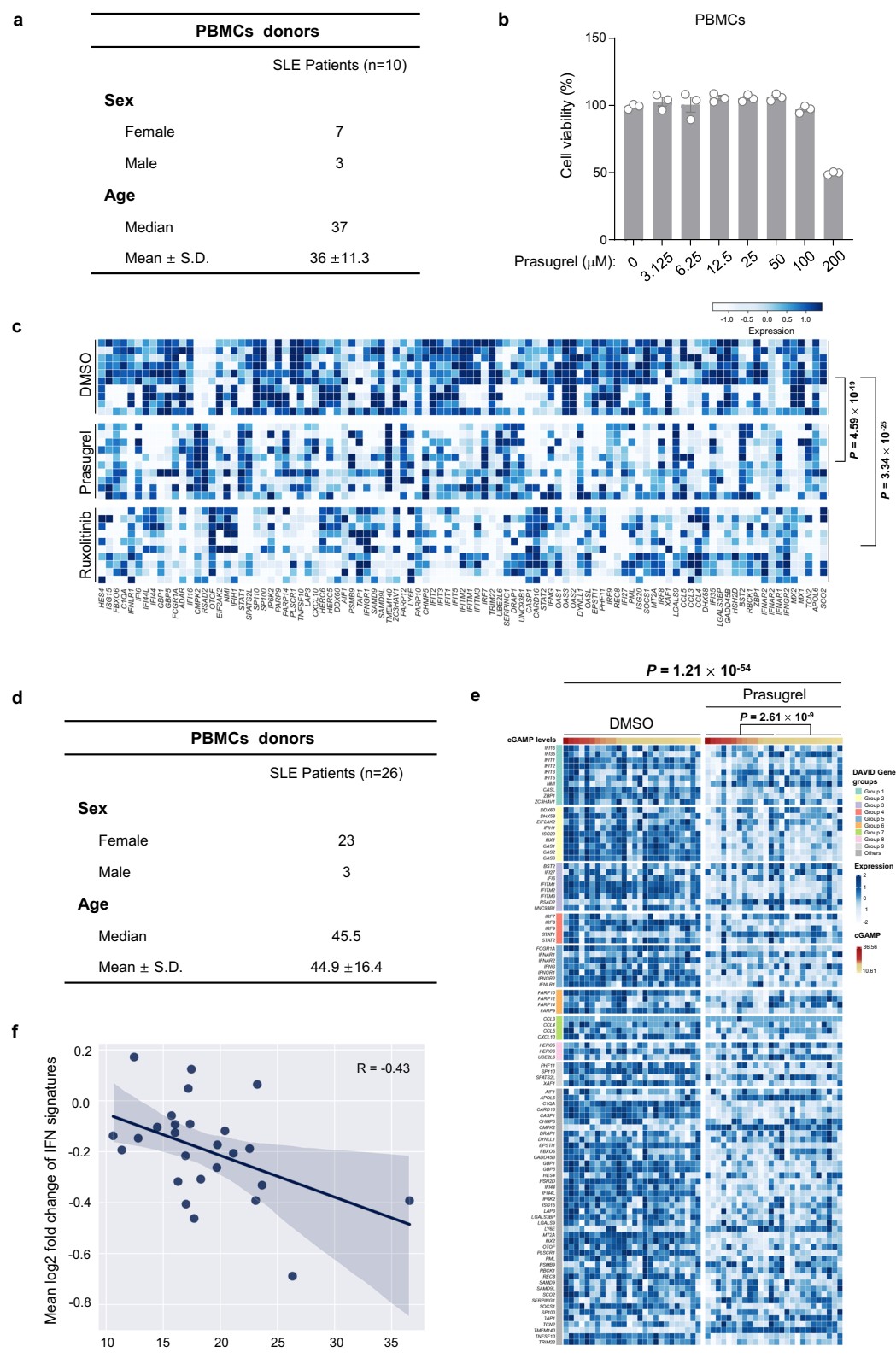

675 → 524; m/z 675 → 136. Levels of cGAMP in cell lysates and plasma were also measured by 2′3′-cGAMP ELISA Kit (501700, Cayman Chemical) according to the manufacturer's instruction.

**Enzyme-linked immunosorbent assay**
Mouse serum was collected and measured for different readouts, such as anti-nuclear antibody (ANA), anti-dsDNA antibody and IFN-α. These above readouts were analyzed with mouse ELISA kits (JL17316, Mouse ANA ELISA kit; JL37836, Mouse anti-dsDNA ELISA kit; EMC035a.96, Mouse IFN-α ELISA kit) according to the manufacturer's instructions. The supernatants of U937 cell with HT-DNA or poly(I:C) stimulation were collected and measured for IFN-β concentration using a human IFN-β ELISA kit (EHC026b.96, Neobioscience) according to the manufacturer's instructions.

**Fig. 6 | Prasugrel exhibits potential therapeutic effect in SLE patient cells.**
**a** Table of information of SLE patients (*n* = 10). **b** PBMCs were incubated with prasugrel at indicated concentrations for 24 h. Cell viabilities were assessed using MTS assay. Data are presented as the means ± SEM from three independent experiments. **c** Heatmaps showing relative expression of genes of IFN signature in SLE patient PBMCs treated with DMSO, prasugrel or ruxolitinib (5 μM). **d** Table of information of SLE patients (*n* = 26). **e** Heatmaps showing relative expression of genes of IFN signature in SLE patient PBMCs treated with DMSO or prasugrel (Samples are sorted in descending order based on their cGAMP levels). Genes were ordered in functional similarity groups defined by DAVID gene functional classification. Genes not assigned to any functional similarity group were categorized as 'Others'. **f** Association between cGAMP levels and mean log2-transformed fold change of expression levels of IFN signature genes quantified using Pearson's correlation coefficient. The solid line indicates the linear regression fit. The error bands represent the 95% confidence intervals. Statistical analysis was performed using two-tailed student's paired *t*-test. P values were corrected using Holm-Bonferroni method and are shown in figures. IFN signature gene list (**c**, **e**, **f**, *Ref.* *PMID: 32747814*). Source data are provided as a Source Data file.

## Tissue staining and immunofluorescence

Kidneys were fixed in 4% PFA (P1110, MACGENE) for 24 h and then dehydrated overnight. The dehydrated samples were embedded in paraffin and sectioned at a thickness of 4 μm. For HE staining, the sections were stained with hematoxylin and eosin. After drying, the sections were mounted and scanned on NDP.view2. For immunofluorescence staining, the sections were deparaffinized in xylene, serial solutions of alcohol and water, and then were performed antigen retrieval by steaming in citric acid. After cooling to room temperature, the sections were washed with PBS twice and blocked in PBS containing 5% normal goat serum (NGS, ZLI-9056, ZSGB-BIO) and 0.3% Triton X-100 (V900502, Sigma-Aldrich). Next, the tissue regions were incubated with AF488-conjunted IgM antibody or AF647-conjunted IgG antibody at 4 °C for overnight. On the second day, the sections were washed with PBS twice and then were stained with Hoechst 33342 (H3570, Thermo Fisher Scientific) for 15 min. Finally, the sections were mounted and visualized on the SLIDEVIEW VS200 slide scanner (Olympus).

## Detection of mtDNA in cytosolic extracts

Bone marrow cells were treated with 5 μg/mL R837 or R848 for indicated times. Cytosolic fractions were obtained by bone marrow cells with 10 μg/mL digitonin (D141, Sigma-Aldrich) in isolation buffer (AAPR406, PythonBio) for 10 min on ice, then centrifuged at 1000 *g* for 5 min at 4 °C. The nuclear DNA extracted from the pellet cells served as the normalization control. The cytosolic supernatants were transferred to fresh tubes and spun at 15,000 *g* for 10 min at 4 °C. MtDNA of supernatants was measured by qPCR.

## Calcein-quenching assay

Bone marrow cells were treated with 5 μg/mL R837 or R848 for 6 h, followed by the cells collection and stained with Fixable Viability Dye eFluo 780 (65-0865-14, Thermo Fisher Scientific) in media with 1% serum for 15 min. Then, the cells were stained with Mitochondrial Permeability Transition Pore Assay Kit (40756ES60, YEASEN) according to the manufacturer's instruction. Cells were sorted and analyzed on a Beckman CytoFlex using CytExpert (Beckman Coulter).

## Purification of site-specific acetylated cGAS recombinant proteins

The site-specific acetylated cGAS recombinant proteins were purified as previous described. Briefly, the pCDF-cGAS$^{K384Ac}$, pCDF-cGAS$^{K394Ac}$ and pCDF-cGAS$^{K414Ac}$ were individually transformed with pAcKRS-3 into BL21 (DE3) cells. The cells were cultured overnight in LB medium containing 50 μg/mL kanamycin and 50 μg/mL spectinomycin. 2 mL cultured bacteria were transferred into 200 mL fresh LB medium containing 50 μg/mL kanamycin, 50 μg/mL spectinomycin and 2 mM acetyl-lysine for further culturing at 37 °C, 220 rpm. When the OD$_{600}$ reached 0.6, 30 mM NAM and 20 mM acetyl-lysine were added and incubated with the bacteria for 20–30 min at 37 °C, 220 rpm. 0.2 mM isopropyl-β-D-1-thiogalactopyranoside (IPTG) (VA20321, GenStar) was then added and the bacteria were cultured for 16–20 h to express the cGAS proteins at 16 °C, 180 rpm. The cells were harvested and resuspended in lysis buffer (20 mM Na$_3$PO$_4$, 1.5 M NaCl, 20 mM imidazole, pH7.5). Following the sonication and centrifugation, the cleared supernatants were purified with His-tag Purification Resin (P2218, Beyotime) according to the manufacturer's protocol.

## In vitro phase separation assay

All the assays were performed in 1 × PBS (137 mM NaCl, 2.7 mM KCl, 10 mM Na$_2$HPO$_4$, 1.8 mM KH$_2$PO$_4$, pH7.4). Recombinant cGAS protein was diluted to a final concentration of 10 μM in PBS. Subsequently, 4 μM FAM-labeled dsDNAs or 5% PEG-8000 were added to the protein solutions. The reaction mixture was transferred to a glass-bottom culture dish after gently mixed, and the droplet formation was visualized by microscopy at the indicated time points. For assays involving inhibitors pre-treatment, cGAS was incubated with prasugrel (5 μM), XQ2B (10 μM), or G140 (10 μM) for 1 h. The reaction buffer was exchanged for PBS prior to performing the LLPS assay. FAM-labeled dsDNAs were prepared from equimolar amounts of the sense and antisense DNA oligonucleotide (sense strand sequence: 5′-ATGGG-CAAAGGAGATCCTAAGAAGCCGAGAGG CAAAATGTCATCATATGCAT TTTTTGTG-3′).

## Time-lapse and Immunofluorescence of cells

H1299-cGAS-mEGFP cells were seeded in a glass bottom cell culture dish and cultured overnight, followed by the treatment of prasugrel (100 μM) for 24 h or XQ2B (10 μM) for 3 h. For time-lapse, the cells were then transfected with 2 μg/mL HT-DNA and the images were acquired by DeltaVision deconvolution microscope at 5-min intervals. For immunofluorescence, the cells were transfected with 4 μg/mL Cy3-DNA and fixed with 4% paraformaldehyde for 15 min, permeabilized with 0.3% Triton X-100 for 10 min and blocked in 5% NGS for 1 h. Cells were incubated with anti-G3BP1 antibody for 1 h at room temperature. Images were acquired using ZEISS LSM 900 Confocal Microscopy. Cy3-DNA was prepared from equimolar amounts of the sense and antisense DNA oligonucleotide (sense strand sequence: 5′-ATGGGCAAAGGAGATCCTAAGAAGCCGAGAGG CAAAATGTCATCA-TATGCATTTTTTGTG-3′).

## RNA sequencing

Total RNA samples were assessed for purity and integrity using a NanoDrop 2000 spectrophotometer and a LabChip GX Touch system, respectively. Eukaryotic mRNA was enriched from total RNA using Oligo (dT) magnetic beads, and sequencing libraries were prepared using the VAHTS Universal V6 RNA-seq Library Prep Kit for Illumina (NR604, Vazyme) according to the manufacturer's instructions. Sequencing of 2 × 150 bp paired-end reads was performed on Illumina NovaSeq X Plus platform.

## RNA-seq data analysis

Quality control and pre-processing of the RNA-seq data were performed using fastp (v0.23.2). Clean reads were mapped to human reference genome GRCh38 using HISAT2 (v2.2.1). Gene expression levels were determined using featureCounts (v2.0.1), and were normalized using DESeq2 (v1.38.3). To facilitate comparison between prasugrel-treated and DMSO-treated PBMC samples from each SLE patient, we employed a gene-wise normalization strategy on paired samples from each individual. For each gene within each patient, the expression levels in the prasugrel-treated and corresponding DMSO-

treated samples were scaled so that the mean expression of that gene across the pair was set to 1. This was achieved by dividing each gene's expression value by the average expression of that gene across the two samples. This approach aims to adjusts for gene-specific expression variability within each patient and accounts for inter-individual heterogeneity, allowing for direct comparison of expression changes induced by prasugrel treatment. For heatmap visualization, genes were first clustered into functional similarity groups using DAVID gene functional classification and were ordered based on these groups[58]. Z-score normalization was performed on the expression levels of each gene across all samples. The z-scores were calculated by subtracting the mean expression of each gene and dividing by the standard deviation, resulting in standardized expression values with a mean of 0 and a standard deviation of 1. For comparison of DMSO-treated, prasugrel-treated and JAK inhibitor-treated PBMC samples from each SLE patient, the similar gene-wise normalization strategy was performed. Specifically, for each gene within each patient, the expression levels in the DMSO-treated, prasugrel-treated and JAK inhibitor-treated samples were scaled so that the mean expression of that gene across the three samples was set to 1. Z-score normalization was then performed on the expression levels of each gene across all samples.

### Statistical analysis

GraphPad Prism (version 10.1) was used for statistical analysis. A standard two-tailed unpaired Student's *t* test was used for statistical analysis of two groups. Statistical analyses from experiments with multiple groups were performed with one-way or two-way ANOVA with Turkey comparisons test. Survival curve comparison was performed with Log-rank (Mantel-Cox) test. A two-tailed paired student's *t* test was used for statistical analysis of IFN signature expression differences between DMSO- and prasugrel-treated PBMC samples and of summed IFN signature expression differences between DMSO-, prasugrel- and JAK inhibitor-treated PBMC samples. Association between cGAMP levels and mean log2-transformed fold change of expression levels of IFN signature genes quantified using Pearson's correlation coefficient. The error bands represent the 95% confidence intervals. $P < 0.05$ were considered as statistically significant.

### Reporting summary

Further information on research design is available in the Nature Portfolio Reporting Summary linked to this article.

## Data availability

The RNA-sequencing data reported in this paper have been deposited in the Genome Sequence Archive for Human (GSA-Human) under the accession code PRJCA030851. All data are included in the Supplementary Information or available from the authors, as are unique reagents used in this Article. The raw data for charts and graphs are available in the Source Data file. Source data are provided with this paper.

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

## Acknowledgements

We sincerely thank Prof. Zheng-gang Liu for illuminating discussions. We are grateful to the Jiaxing Central Blood Station for efforts to recruit healthy participants. This work was supported by grants from China National Natural Science Foundation (82130052 and 81925017 to Tao Li; 32100421 to Shuai Jiang; 82302491 to Ming Zhao).

## Author contributions

T. L., X.-M.Z., and X.-H.H. supervised the project. T.L., X.-H.H., Z.-L.G., L.-M.S., S.J., and M.Z. designed the experiments and performed data analyses. Z.-L.G., L.-M.S., S.-Z.J., L.-H.Y., and Z.-X.L. performed the mouse experiments. Z.-L.G., L.-M.S., M.Z., Ying Yuan, W.X., S.-C.Y., X.-X.Y., P.-P.Z., and Yu Yu performed the experiments in cells, with help from Q.-Y.H., Z.-H.S., X.-P.Y., H.C., T.X., J.-Q.W., and S.J. performed the bioinformatic analyses. X.-C.L. performed the docking simulations. C.-M.W. performed the evaluation of the pathological sections. K.W. and X.X. collected and analyzed immunofluorescence sample data. T.L., Y.-H.L., J.-J.Q., Y.-K.F., Z.-G.L., and Q.F. collected clinical samples. Y.C., W.-H.L., X.-M.Z., A.-L.L., and T.Z. provided guidance for the experiments and analyzed data. T.L., Z.-L.G., L.-M.S., S.J., M.Z., X.-H.H., and Q.F. wrote and revised the manuscript.

## Competing interests

The authors declare no competing interests.
