## [Transparent Peer Review file · Nature Communications]

Prasugrel inhibits TLR7-driven autoimmunity in systemic lupus erythematosus by acetylating cGAS

Corresponding Author: Dr Tao Li

Version 0:

Reviewer comments:

Reviewer #1

(Remarks to the Author)

The manuscript by Guo et al presents a novel cGAS inhibitor, prasugrel, which they identified by in silico screening of 3,159 FD approved drugs. They initially present data to justify the targeting of cGAS – looking at cGAMP levels in SLE plasma and showing a positive correlation between TLR7 (an important driver of SLE) and cGAS. They next show that in the TLR7 imiquimod model of inducible lupus that Imiquimod induces an early and sustainable level of cGAMP in mice (measured in plasma) and that loss of cGAS ameliorates lupus in this model, thus providing a strong rationale for targeting cGAS in SLE. Having identified prasugrel, they show that it acetylates cGAS on lysines 384, 394 and 414 and in doing so prevents its phase separation and mediates its activity. They use Trex1^{-/-} cells and mice to show prasugrel rescues pathology and the imiquimod model to show it protects in an SLE model of disease. Using SLE patients cells they also demonstrate that it reduces ISG expression. Importantly their data shows that prasugrel has a 40 fold higher potency than aspirin, another known drug that acetylates cGAS. This is a well presented and nicely controlled study. Although the role of acetylation of cGAS in regulating its activity is not particularly novel, the identification and characterization of prasugrel as a novel cGAS inhibitor is an important observation. The data presented initially showing that TLR7 can drive cGAMP induction in vivo is intriguing and very interesting. However, the correlations described in figure 1 and extended data showing an association with TLR7 and cGAS cannot be interpreted as showing association as cGAS is IFN inducible and this association most likely reflects this. The experiments showing that acetylation of cGAS at the specific lysines (K384/394/414) disrupts DNA-induced liquid phase condensates of cGAS suggest this is the mechanism behind prasugrel's activity.

In summary, this is a well conceived study delivering important new insights into crosstalk between TLR7 and cytosolic nucleic acid sensing albeit without the details. It also demonstrates that acetylation of cGAS on these 3 residues reduces the formation of DNA:cGAS LLS condensates, presumably resulting in decreased function. A number of minor points have been raised above. Additional points are:

The inclusion of polyI:C as a control in figure 2 is appropriate and a good control, however, the language used to describe its use in the main body of the text needs to be clearer.

Also minor typographical errors were spotted – one including re. 32.

Reviewer #2

(Remarks to the Author)

The paper by Guo et al. reports that the antiplatelet drug prasugrel inhibits cGAS activity by preventing DNA-induced cGAS condensation. They also demonstrate that cGAS is significantly activated in SLE patients, as evidenced by elevated cGAMP levels in serum, and that cGAS deletion safeguards mice against lupus induced by TLR7 activation. They also demonstrate that the in vivo administration of prasugrel in TREX1-deficient mice alleviates lupus symptoms and that the treatment of whole blood from SLE patients reduces the expression of interferon-stimulated genes. The paper is well written, the findings are interesting, and they are potentially of important biomedical relevance. However, several experimental studies are needed to support the conclusions.

Specific comments

1. Line 54: The deficiency of TREX1 has been identified in patients with SLE 22 and 54 Aicardi-Goutières syndrome (AGS) 23-25. Instead of reference 22, the paper by Lee-Kirsch et al. (Nat Genet. 2007 Sep;39(9):1065-7. doi: 10.1038/ng2091) should be cited.

2. cGAMP levels were found to be increased in SLE patients, but not in RA or DM patients. Like SLE patients, DN patients exhibit exquisitely strong type I IFN activation, yet apparently no elevated cGAMP levels. Were the DM patients studied in Figure 1 tested for ISG expression? How do the authors reconcile this discrepancy?

3. cGAMP levels and cGAS expression were found to correlate with TLR7 gene expression. The authors argue that TLR7 hyperactivity drives lupus. However, this correlation was weak ($R = 0.46$). As both cGAS and TLR7 are ISGs, this correlation is not surprising and does not imply a causal relationship. Similarly, reduced R837-induced IFN- α production in cGAS-deficient mice or increased cGAS activity in cells stimulated with R837 likely results from the general negative or positive effects of alerting the IFN axis, which has an effect on ISGs such as TLR7 and cGAS.

4. The authors found that prasugrel specifically acetylates cGAS at K384, K394 and K414. Previous studies had shown that this process inactivates cGAS. They also show that prasugrel specifically inhibits DNA-induced IFN production in BMDMs. To demonstrate that this is dependent on cGAS acetylation, the authors should test DNA-induced IFN production in BMDMs or mice, in which the cGAS acetylation sites have been mutated.

5. The authors do not provide a detailed description of the experimental set-up for in vitro condensation of cGAS (buffer, pH, concentrations). This makes it difficult to assess the validity of the experiments. The authors need to include positive and negative controls.

6. cGAS-mEGFP-expressing cells showed reduced foci formation upon prasugrel treatment. The authors should stain these foci for G3BP1 and DNA to confirm condensate formation. The authors should use known inhibitors of phase separation as control.

7. In addition to the *trex1*-deficient mouse, the authors should validate their findings in pristane-induced SLE mice and MRL/lpr mice, which are bona fide mouse models of complex SLE, by assessing cGAS activation (cGAMP levels) and by testing the therapeutic effect of prasugrel. What was the effect of prasugrel treatment of *trex1*-deficient mice on ISG expression and on B cells?

8. For comparison of the IFN-inhibitory potential of prasugrel, the authors should also treat whole blood from SLE patients with other inhibitors of the IFN axis, such as JAK inhibitors, anifrolumab or TLR7 antagonists. Presumably, cGAS hyperactivity will only be the primary disease-causing factor in a fraction of patients. It is surprising to observe such a profound effect of prasugrel in all patients. What were the clinical characteristics of these patients? Did the authors identify any variables that could be used for patient stratification?

Reviewer #3

(Remarks to the Author)

This manuscript presents interesting results that support the thesis that an anti-platelet drug, prasugrel, already used in the clinic as anti-clotting treatment, can also inhibit cGAS activation by acetylation of specific lysines. The same group published on Cell 5 years ago that cGAS activation can be inhibited through lysine acetylation by aspirin. Here the authors show that prasugrel can do what aspirin can do at a lower dose. Since the high doses of aspirin required to inhibit cGAS would have many side effects in patients, having found a different drug that carry less side effects is very significant.

The paper also shows results supporting the thesis that prasugrel can treat lupus autoimmunity, shown in two models. One model is cGAS dependent, the *Trex*^{-/-} mice, and one is TLR7 dependent, Balb/c mice painted on the skin with imiquimod. The inhibition by prasugrel of the autoimmunity in *Trex1*^{-/-} mice is clear and convincing, suggesting a novel treatment in the cGAS dependent interferonopathies, like Aicardi-Gutierrez. The results of the effects of prasugrel on Imiquimod autoimmunity are also clear for the part of the splenomegaly and the expression of I-IFNs and autoantibodies, but the data on the glomerulonephritis are not strong (see below).

Moreover, the manuscript proposes that cGAS is downstream of and contributes to TLR7 activation and TLR7-dependent autoimmunity by imiquimod.

The lack of some important controls weakens the rigor of the contribution of cGAS to TLR7 activation and autoimmunity. In details:

The authors start the manuscript with the hypothesis that cGAMP, product of cGAS activation, is higher in the plasma of SLE patients because is induced by TLR7 activation, which is high because the TLR7 expression is higher in SLE patients than healthy controls. These results could be explained in many other ways. For example, cGAMP and TLR7 RNA expression could correlate because higher cGAMP could be induced by some form of DNA through the activation of cGAS and STING and the production of type I-IFNs, and the latter has been shown to increase TLR7 expression. Therefore, TLR7 expression could be the effect rather than the cause of the high levels of cGAMP in SLE.

Fig.1c-d are important to support a direct induction of cGAS activation by TLR7 stimuli, but the legend does not clarify whether the three dots per each bar are technical replicates from one culture or from three individual cultures from three mice. More repeats of the experiments are required to support the rigor of these figures.

Moreover, there is no justification for using a mix population of bone marrow precursors instead of BMDMs or BMDCs to test TLR7 responses. Furthermore, IFN α is mostly produced by plasmacytoid dendritic cells (pDCs) and therefore testing the

production of IFN α in cGAS $^{-/-}$ pDCs would provide stronger data to support the dependence of the IFN α production on cGAS. Alternatively, the authors could measure IFN β produced by BMDMs and BMDCs.

The conclusion that R837/Imiquimod induces cGAS activation and production of cGAMP is a very novel one that requires stronger results. It would be important to provide the control of stimulating with R837 and measuring cGAMP production in cells that are both TLR7 $^{-/-}$ and TLR7 $^{-/-}$ /cGAS $^{-/-}$, to exclude the possibility that R837 induces cGAMP in a TLR7 independent manner. Moreover, would other TLR7 ligands induce the same cGAMP production or is it something specific of imiquimod? Does it require TLR7?

The role of cGAS in the Imiquimod-induced lupus is still unclear. Fig. 1m shows the staining for IgM in murine glomeruli as sign of lupus disease, but it is standard procedure to show the staining for IgG in the glomeruli, since IgG are the sign of a memory response against the kidney, as it is regularly observed in autoimmune mice.

The histology of the Imiquimod kidney presented in Fig. 1n does not show the standard features of glomerulonephritis that are usually present in mice developing lupus, including the imiquimod type (PMID 35297032, 35260898). It is recommended that the H&E slides of the kidney are read by an expert in murine lupus nephritis. Same comments for Figure 5g and h. The potential of prasugrel as novel therapeutic strategy in human SLE is overstated. The results in Fig. 6 indicate that prasugrel decreases the IFN Signature in cells from SLE patients. These effects are not direct indicators that prasugrel will be therapeutic in SLE since anifrolumab, which decreases the IFN Signature, has only mild effects on lupus disease. Moreover, Figure 6F reports the meta-analyses of public data that indicate that the expression of the IFN Signature inversely correlates with the cGAMP levels in the plasma, but it is not clear what is the source of the measurements of the cGAMP levels (the cited papers?) and how could serve as a predictive biomarker to identify SLE patients who are likely to respond favorably to prasugrel treatment.

Methods

Statistical significance was assessed using two-tailed student's unpaired t-test even for data in which 3 or more conditions were analyzed, which would require ANOVA and post-hoc analyses.

The authors state that antibodies against cGAS, Acetylated-cGAS (K384+K394) and Ac-cGAS (K414) were generated in their laboratory, but it is not mentioned whether those Antibodies had been validated before and how. This validation is crucial. If it was done in a previous publication, please say so and cite the paper.

Minor points

Please spell out what AcK means.

The following sentence infers a translational potential in the results of Figure 6d-e and Extended Data Fig.3c,d that goes beyond the effective potential of those data. "Because the systemic disease characteristics of SLE were unable to evaluate in patient cells, we used several published IFN signatures^{30,37-39} to reflect the autoimmune status in these cells. Significantly, prasugrel ameliorated the autoimmune status in patient cells, as indicated by the expression change of IFN signatures (Fig.6d,e and Extended Data Fig.3c,d). The mentioned figures instead show an inverse correlation between the levels of cGAMP and those of the IFN Signature. Any comment on the "autoimmune status" is unjustified and not necessary to support the relevance of the paper. Please change the wording of these sentences.

Version 1:

Reviewer comments:

Reviewer #1

(Remarks to the Author)

The authors have addressed all my concerns and the additional data requested by reviewers 2&3 have strengthened the findings. I recommend acceptance

Reviewer #3

(Remarks to the Author)

The authors of this paper have answered to the critiques with great effort. I especially appreciate the openness to show raw data, like the full kidney histology, and the large number of unpublished results for the reviewers's eye only. These data have dissipated past concerns. Moreover, they discovered the mechanism that link TLR7 and cGAS, making this paper very novel indeed.

Without asking new experiments, I have few suggestions to improve the clarity of the manuscript.

1. The results of the inhibitors of M5049 and Ru.521 showed in Figure R11 and R12 are very interesting and strongly support the paper. Can they be added to this manuscript?
2. The results of R5-, R6, R7 are also very important (R7 above all) to put in perspective the conclusion of this paper with previous papers suggesting that cGAS is not important in lupus. It would be great to add them to this paper. If the authors are already planning to put them in another paper, maybe they can mention them in the discussion as data in a manuscript in preparation.
3. Regarding the histology of the kidney in Balb/c mice treated with Imiquimod, Figure R13 shows that this model does not

have the classic signs of lupus glomerulonephritis, but rather immune infiltrates in the lower tubular and medullary areas, and those infiltrates do not form with prasugrel treatment or in *cgas*^{-/-} mice. Therefore, figure R13 strongly supports the conclusions of the paper and would be helpful to show it. Moreover, it would remind the reader that Balb/c or in general WT mice have mild forms of kidney disease that are not exactly lupus nephritis. Showing R13 in Supplemental would NOT diminish the strength of the conclusions and would be didactic for the readers, reminding them of the limitations of the murine models with work with.

4. Figure 6 supports the conclusion that prasugrel inhibits the expression of genes linked to cGAS activation and inflammation in general, and it has more effect in SLE patients with high levels of cGAMP. In Fig. 6e the general effect of prasugrel is clear with a naked eye, but the link between the levels of cGAMP and the effects of prasugrel is not that evident. After spending some time on the Figure, I am convinced but it could help the readers if the genes, rather than in alphabetical order, were clustered in functional groups, like ISGs, vs. chemokines, vs. other groups and etc, since it seems that they behave similarly inside groups, facilitating the recognition of trends and behaviors with the eye.

Point-by-Point Response:

Reviewer #1:

The manuscript by Guo et al presents a novel cGAS inhibitor, prasugrel, which they identified by in silico screening of 3,159 FDA approved drugs. They initially present data to justify the targeting of cGAS – looking at cGAMP levels in SLE plasma and showing a positive correlation between TLR7 (an important driver of SLE) and cGAS. They next show that in the TLR7 imiquimod model of inducible lupus that Imiquimod induces an early and sustainable level of cGAMP in mice (measured in plasma) and that loss of cGAS ameliorates lupus in this model, thus providing a strong rationale for targeting cGAS in SLE. Having identified prasugrel, they show that it acetylates cGAS on lysines 384, 394 and 414 and in doing so prevents its phase separation and mediates its activity. They use Trex1-/- cells and mice to show prasugrel rescues pathology and the imiquimod model to show it protects in an SLE model of disease. Using SLE patients cells they also demonstrate that it reduces ISG expression. Importantly their data shows that prasugrel has a 40-fold higher potency than aspirin, another known drug that acetylates cGAS. This is a well presented and nicely controlled study. Although the role of acetylation of cGAS in regulating its activity is not particularly novel, the identification and characterization of prasugrel as a novel cGAS inhibitor is an important observation. The data presented initially showing that TLR7 can drive cGAMP induction in vivo is intriguing and very interesting. However, the correlations described in figure 1 and extended data showing an association with TLR7 and cGAS cannot be interpreted as showing association as cGAS is IFN inducible and this association most likely reflects this. The experiments showing that acetylation of cGAS at the specific lysines (K384/394/414) disrupts DNA-induced liquid phase condensates of cGAS suggest this is the mechanism behind prasugrel's activity.

In summary, this is a well conceived study delivering important new insights into crosstalk between TLR7 and cytosolic nucleic acid sensing albeit without the details. It also demonstrates that acetylation of cGAS on these 3 residues reduces the formation of DNA:cGAS LLS condensates, presumably resulting in decreased function. A number of minor points have been raised above.

Response: The reviewer indicated that our work is a well presented and nicely controlled study. We greatly appreciate the reviewer's encouraging comments. Following the reviewer's suggestions, we revised our manuscript accordingly.

1. Regarding the reviewer mentioned points: “Correlations described in figure 1 and extended data showing an association with TLR7 and cGAS cannot be interpreted as showing association, as cGAS is IFN inducible and this association most likely reflects this” and “this is a well conceived study delivering important new insights into crosstalk between TLR7 and cytosolic nucleic acid sensing albeit without the details”.

Response: We thank the reviewers for the insightful comments. We agree with the reviewer that both *TLR7* and *cGAS* are interferon-stimulated genes (ISGs), and thus the observed coordination in their expression is expected. Reviewer#2 and #3 also raised similar points. We apologize for our unclear descriptions.

According to previous publications, *TLR7* is proposed as a key driver of lupus in humans. For example, a functional polymorphism of *TLR7* was identified predisposing to SLE in humans (*PNAS*, 2010, PMID: 20733074). The unrestricted *TLR7* signaling is associated with human lupus (*Science Immunology*, 2024, PMID: 38207015). Moreover, a gain-of-function mutation in *TLR7* (Y264H) was reported to cause human SLE (*Nature*, 2022, PMID: 35477763). In our study, we sought to investigate whether *cGAS* plays a role in *TLR7*-driven SLE. Therefore, we first determined whether *cGAS* is activated in SLE patients and then assessed the correlation between *TLR7* expression and *cGAMP* levels (indicator of *cGAS* activation) (Fig.1b in our revised manuscript). These results may reflect the potential interplay between *cGAS* and *TLR7* signaling.

Using mouse models, we further showed that in the *TLR7*-driven SLE mice, *cGAS* was indeed activated (Fig.1d, 1e in our revised manuscript) and required (Fig.1f-I in our revised manuscript). We also confirmed that *TLR7*-stimulated IFN- α production was markedly dampened in the absence of *cGAS* (Fig.1m, 1n and Supplementary Fig.11, 1n in our revised manuscript). As suggested by Reviewer#2, we used ruxolitinib, a JAK1/2 inhibitor (*The New England journal of medicine*, 2012, PMID:22375970), to block the IFN cascade downstream of *TLR7*. Under such condition, *TLR7* agonist stimuli still effectively triggered the synthesis of *cGAMP* (Fig.1o in our revised manuscript), suggesting that the involvement of *cGAS* in *TLR7* signaling is independent of the feedback of IFN axis. Thus, *cGAS* may regulate the activation of *TLR7* signaling.

We further studied the role of *cGAS* in *TLR7* signaling. We found that *TLR7* activation led to the increased mitochondrial permeability and the cytosol mtDNA releasement (Supplementary Fig.2a-c in our revised manuscript). The released mtDNAs stimulated the activation of *cGAS*, which is important for the effective production of IFN- α downstream *TLR7* (Fig. 1c, 1m, 1n and Supplementary

Fig.1c, 1l, 1n in our revised manuscript). We further show that the mtDNAs are released *via* BAX and BAK macropores, as the BAK/BAX oligomerization inhibitor MSN-125 (*Cell Chemical Biology* 2017, PMID: 28392146) robustly suppressed the mtDNA release induced by TLR7 activation (**Supplementary Fig.2d-f** in our revised manuscript).

Collectively, our new data demonstrate that cGAS is essential for the TLR7-driven IFN- α production. With these new data, we uncovered the mechanism underlying the role of cGAS in TLR7-signaling and provided important new insights into crosstalk between TLR7 and cytosolic DNA sensing. With the adding of these new data, our manuscript is significantly improved. In the revised manuscript, we rephrased our rationale for studying TLR7-driven SLE and removed data regarding the correlation between cGAS mRNA levels and *TLR7* expression.

Additional Points:

2. *The inclusion of polyI:C as a control in figure 2 is appropriate and a good control, however, the language used to describe its use in the main body of the text needs to be clearer.*

Response: We thank the reviewer for this suggestion and revised our manuscript to add detailed descriptions of poly(I:C). We also explained why we used poly(I:C) as control (**Line.143 – Line.146** in our revised manuscript).

3. *Also minor typographical errors were spotted – one including re. 32.*

Response: We thank the reviewer for the suggestion. Accordingly, we corrected the reference citation style and checked our revised manuscript carefully.

Reviewer #2:

The paper by Guo et al. reports that the antiplatelet drug prasugrel inhibits cGAS activity by preventing DNA-induced cGAS condensation. They also demonstrate that cGAS is significantly activated in SLE patients, as evidenced by elevated cGAMP levels in serum, and that cGAS deletion safeguards mice against lupus induced by TLR7 activation. They also demonstrate that the in vivo administration of prasugrel in TREX1-deficient mice alleviates lupus symptoms and that the treatment of whole blood from SLE patients reduces the expression of interferon-stimulated genes. The paper is well written, the findings are interesting, and they are potentially of important biomedical relevance. However, several experimental studies are needed to support the conclusions.

Response: The reviewer indicated that our study presents interesting findings. We greatly appreciate the reviewer's encouraging comments. We also thank the reviewer for his/her important suggestions to further improve our work. Following the reviewer's suggestions, we conducted substantial experiments and revised our manuscript extensively. As detailed below, point-by-point, we can address all concerns with new data and discussions.

Specific Comments:

1. *Line 54: The deficiency of TREX1 has been identified in patients with SLE (22 and 54) Aicardi-Goutières syndrome (AGS) (23-25). Instead of reference 22, the paper by Lee-Kirsch et al. (Nat Genet. 2007 Sep;39(9):1065-7. doi: 10.1038/ng2091) should be cited.*

Response: Following the reviewer's suggestion, we replaced the original reference 22 with the paper by Lee-Kirsch *et al.* (Nat Genet. 2007 Sep;39(9):1065-7. doi: 10.1038/ng2091) in our revised manuscript (**Ref # 23**).

2. *cGAMP levels were found to be increased in SLE patients, but not in RA or DM patients. Like SLE patients, DM patients exhibit exquisitely strong type I IFN activation, yet apparently no elevated cGAMP levels. Were the DM patients studied in Figure 1 tested for ISG expression? How do the authors reconcile this discrepancy?*

Response: We thank the reviewer for this important point. To address this point, we newly collected peripheral blood samples from 6 DM patients and 10 SLE patients, and measured the plasma cGAMP levels, in line with our data in the original manuscript, the cGAMP levels in DM patients were not significantly

elevated compared to that of SLE patients (Figure R_1).

We also determined the expression of ISGs in PBMCs of DM patients and healthy controls using RNA-seq. Our new data showed that the DM patients exhibit exquisitely strong type I IFN activation (Figure R_2). This is also consistent with previous studies (*Current Rheumatology Reports*, 2011, PMID: 20425524; *Arthritis & Rheumatology*, 2023, PMID: 37096447).

Figure R_1. Plasma cGAMP concentrations of healthy donors (n = 6), SLE patients (n = 10) and DM patients (n = 6). Data were analyzed by one-way ANOVA with Tukey's multiple comparisons test.

Figure R_2. The expression of ISGs in healthy donors (n = 6) and DM patients (n = 6). Data were analyzed by unpaired Student's *t*-test.

According to previous publications, the type I IFN expression in DM is likely triggered by the RIG-I-like receptors (RLRs), such as RIG-I and MDA-5, which are activated upon the detection of intracellular RNAs (*Journal of Pathology*, 2014, PMID: 24604766; *Nature Reviews Rheumatology*, 2024, PMID: 38057474; *Arthritis Research & Therapy*, 2019, PMID: 31142372; *Arthritis Research & Therapy*, 2017, PMID: 28738907). As shown in **Figure R_3** below, the RNA mimics poly(I:C) triggers dramatic transcription of type I IFN, but not the synthesis of cGAMP. This may explain why the cGAMP levels, indicator of the activation of intracellular DNA sensor cGAS, were not elevated in DM patients.

3. *cGAMP* levels and *cGAS* expression were found to correlate with *TLR7* gene expression. The authors argue that *TLR7* hyperactivity drives lupus. However, this correlation was weak ($R = 0.46$). As both *cGAS* and *TLR7* are ISGs, this correlation is not surprising and does not imply a causal relationship. Similarly, reduced R837-induced IFN- α production in *cGAS*-deficient mice or increased *cGAS* activity in cells stimulated with R837 likely results from the general negative or positive effects of alerting the IFN axis, which has an effect on ISGs such as *TLR7* and *cGAS*.

Response: We thank the reviewers for the insightful comments. We agree with the reviewer that both *TLR7* and *cGAS* are interferon-stimulated genes (ISGs), and thus the observed coordination in their expression is expected. Reviewer#1 and #3 also raised similar points. We apologize for our unclear descriptions. According to previous publications, *TLR7* is proposed as a key driver of lupus in humans. For example, a functional polymorphism of *TLR7* was identified predisposing to SLE in humans (*PNAS*, 2010, PMID: 20733074). The unrestricted

TLR7 signaling is associated with human lupus (*Science Immunology*, 2024, PMID: 38207015). Moreover, a gain-of-function mutation in TLR7 (Y264H) was reported to cause human SLE (*Nature*, 2022, PMID: 35477763). In light of these important findings, we sought to investigate whether cGAS plays a role in TLR7-driven SLE. Therefore, we first determined whether cGAS is activated in SLE patients and then assessed the correlation between *TLR7* expression and cGAMP levels (indicator of cGAS activation) (**Fig.1b** in our revised manuscript). These results may reflect the potential interplay between cGAS and TLR7 signaling.

Using mouse models, we further showed that in the TLR7-driven SLE mice, cGAS was indeed activated (**Fig.1d, 1e** in our revised manuscript) and required (**Fig.1f-i** in our revised manuscript). We also confirmed that TLR7-stimulated IFN- α production was markedly dampened in the absence of cGAS (**Fig.1m, 1n** and **Supplementary Fig.1i, 1n** in our revised manuscript). Thus, cGAS may regulate the activation of TLR7 signaling.

Following the reviewer's suggestion, we used ruxolitinib, a JAK1/2 inhibitor (*The New England journal of medicine*, 2012, PMID:22375970), to block the IFN cascade downstream of TLR7. Under such condition, TLR7 agonist stimuli still effectively triggered the synthesis of cGAMP (**Fig.1o** in our revised manuscript), suggesting that the involvement of cGAS in TLR7 signaling is independent of the feedback of IFN axis.

We further studied the role of cGAS in TLR7 signaling and found that TLR7 activation led to the increased mitochondrial permeability and the cytosol mtDNA releasement (**Supplementary Fig.2a-c** in our revised manuscript). The released mtDNAs stimulated the activation of cGAS, which is important for the effective production of IFN- α downstream TLR7 (**Fig.1c, 1m, 1n** and **Supplementary Fig.1c, 1i, 1n** in our revised manuscript). We further show that the mtDNAs are released *via* BAX and BAK macropores, as the BAK/BAX oligomerization inhibitor MSN-125 (*Cell Chemical Biology*, 2017, PMID: 28392146) robustly suppressed the mtDNA release induced by TLR7 activation (**Fig.2d-f** in our revised manuscript).

Collectively, our new data demonstrate that cGAS is required for the TLR7-driven IFN- α production. With these new data, we uncovered the mechanism underlying the role of cGAS in TLR7-signaling and provided important new insights into the crosstalk between TLR7 and cytosolic DNA sensing. In the revised manuscript, we rephrased our rationale for studying TLR7-driven SLE and removed data regarding the correlation between cGAS mRNA levels and *TLR7* expression.

4. The authors found that prasugrel specifically acetylates cGAS at K384, K394 and K414. Previous studies had shown that this process inactivates cGAS. They also show that prasugrel specifically inhibits DNA-induced IFN production in BMDMs. To demonstrate that this is dependent on cGAS acetylation, the authors should test DNA-induced IFN production in BMDMs or mice, in which the cGAS acetylation sites have been mutated.

Response: We thank the reviewer for this important suggestion. We agree with the reviewer that using an acetylation-site-mutant cGAS-expressing cell will provide the most compelling evidence. To do so, we will need to mutant all three sites (K384/K394/K414) that can be acetylated by prasugrel. Unfortunately, based on our data, some of the lysine sites are critical for cGAS function and cannot be mutated. For example, we found that K414 is a critical residue for cGAS activity, the mutation of K414 to other amino acids abolished cGAS activity (Figure R_4). Therefore, it is impossible to create a prasugrel-resistant but functional cGAS mutant.

As discussed in our response to Point #7 of this reviewer, prasugrel treatment showed no effect in treating MRL/lpr mice (Figure R_5). This is consistent with the observation that deletion of *Cgas* in this model did not ameliorate autoimmunity (Figure R_6). These data indicate that prasugrel likely suppresses the autoimmune response in mice through inhibiting cGAS.

5. The authors do not provide a detailed description of the experimental set-up for *in vitro* condensation of cGAS (buffer, pH, concentrations). This makes it difficult to assess the validity of the experiments. The authors need to include positive and negative controls.

Response: We thank the reviewer for this point. Following the reviewer's suggestions, we included the following controls.

For liquid-liquid phase separation (LLPS):

- Negative control: cGAS proteins were not incubated with dsDNAs.
- Positive control: we used PEG-8000, a known LLPS inducer (*Nature Communications*, 2023, PMID: 37794023).

Our new results showed that cGAS proteins failed to form condensates in the absence of dsDNAs (**Supplementary Fig.4a** in our revised manuscript). We also used PEG-8000 to promote the condensate formation of wild-type and the site-specifically acetylated cGAS proteins (cGAS^{K384Ac}, cGAS^{K394Ac} and cGAS^{K414Ac}). We find that the unmodified cGAS proteins formed more robust condensates than those of acetylated variants (**Supplementary Fig.4b** in our revised manuscript), indicating that acetylation attenuates the phase separation capacity of cGAS.

We also introduced controls in our study:

- Negative control: G140, an inhibitor that occupies the ATP/GTP-binding site of cGAS (*Nature Communications*, 2019, PMID: 31113940), which has no effect on cGAS LLPS (*Nature Communications*, 2025, PMID: 40461475).
- Positive control: XQ2B, an inhibitor targeting cGAS–DNA binding and blocks cGAS LLPS (*Nature Communications*, 2023, PMID: 37783727).

Our results showed that both prasugrel and XQ2B, but not G140, restrained the formation of cGAS-DNA condensates, while they strongly suppressed cGAS activation (**Fig.3f-h** in our revised manuscript). Together, these results further support the conclusion that acetylation disturbs the formation of DNA-cGAS condensates.

Regarding the experimental design, we apologize for our unclear descriptions in the original manuscript. As suggested by the reviewer, we revised the **Methods** section by adding detailed descriptions of the experimental set-up as follows:

All the assays were performed in 1× PBS (137 mM NaCl, 2.7 mM KCl, 10 mM Na₂HPO₄, 1.8 mM KH₂PO₄, pH7.4). Recombinant cGAS protein was diluted to a final concentration of 10 μM in PBS. Subsequently, 4 μM FAM-labeled dsDNAs or 5% PEG-8000 were added to the protein solutions. The reaction mixture was transferred to a glass-bottom culture dish after gently mixed, and the droplet formation was visualized by microscopy at the indicated time points. For assays involving inhibitors pre-treatment, cGAS was incubated with prasugrel (5 μM), XQ2B (10 μM), or G140 (10 μM) for 1 hour. The reaction buffer

was exchanged for PBS prior to performing the LLPS assay.

6. *cGAS-mEGFP-expressing cells showed reduced foci formation upon prasugrel treatment. The authors should stain these foci for G3BP1 and DNA to confirm condensate formation. The authors should use known inhibitors of phase separation as control.*

Response: Following the reviewer's suggestion, we performed immunofluorescence (IF) staining for G3BP1 and DNA in H1299-cGAS-mEGFP cells to further validate the condensate formation.

H1299-cGAS-mEGFP (green) cells were treated with DMSO or prasugrel for 24 hours, followed by transfection with Cy3-labeled dsDNA (red) to induce cGAS condensation. XQ2B was used as a positive control.

Our results showed a clear colocalization of cGAS with DNA (**Supplementary Fig.4c** in our revised manuscript), which is consistent with our previous findings (*Nature immunology*, 2018, PMID: 30510222; *The EMBO reports*, 2022, PMID: 34779554). Importantly, treatment with prasugrel or XQ2B significantly reduced the formation of these cGAS-DNA droplets (**Supplementary Fig.4c** in our revised manuscript). These findings support a specific role for prasugrel in inhibiting cGAS condensate formation.

7. *In addition to the trex1-deficient mouse, the authors should validate their findings in pristane-induced SLE mice and MRL/lpr mice, which are bona fide mouse models of complex SLE, by assessing cGAS activation (cGAMP levels) and by testing the therapeutic effect of prasugrel. What was the effect of prasugrel treatment of trex1-deficient mice on ISG expression and on B cells?*

Response: We thank the reviewers for the important comments. We agree with the reviewer that pristane-induced mice and MRL/lpr mice are widely used models for SLE study.

According to a previous report (*Frontiers in Immunology*, 2021, PMID: 33854495), cGAS deletion did not alleviate the autoimmunity in these models. In our study we obtained similar results (**Figure R_6**). We also show that prasugrel treatment failed to ameliorate autoimmunity in MRL/lpr mice (**Figure R_5**). Following the reviewer's suggestion, we measured the serum cGAMP levels in different SLE mouse models and found that the elevated serum cGAMP could be detected in IMQ-induced SLE mice, but not in the pristane-induced SLE or MRL/lpr mice (**Figure R_7**). These data indicates that cGAS likely promotes the pathogenesis of SLE in IMQ-induced mouse model.

Figure R_5. ELISA detection of serum IFN- α concentrations, anti-nuclear antibodies and anti-dsDNA antibodies of MRL/*lpr* mice (n = 6) with DMSO or prasugrel treatment. Data were statistically analyzed by two-tailed Student's *t*-test.

Figure R_6. Immunoblot analysis of WT and *Cgas*^{-/-} bone marrow cells from MRL/*lpr* or C57BL/6N mice with indicated antibodies. ELISA detection of serum IFN- α concentrations, anti-nuclear antibodies and anti-dsDNA antibodies of WT (n = 8 to 10) and *Cgas*^{-/-} mice (n = 8 to 10). Data were statistically analyzed by two-tailed Student's *t*-test.

Figure R_7. For IMQ-induced mice and Pristane-induced mice, we examined serum cGAMP levels one week after the initial treatments. 8-10-week-old MRL/*lpr* mice were used to measure the serum cGAMP levels. Data were statistically analyzed by two-tailed Student's *t*-test.

In addition, as suggested by the reviewer, we used the ISG expression levels in the heart tissues of the *Trex1*^{-/-} mice as the indicator for disease severity. Prasugrel significantly reduced ISG expression in the heart tissues (Fig.4f in our revised manuscript).

We further evaluated the effect of prasugrel on B cells of *Trex1*^{-/-} mice. TREX1 deficiency is known to cause aberrant B cell activation and autoantibody production (*Cell* 2008, PMID: 18724932; *European Journal of Immunology* 2022, PMID: 35112355). We thus examined the abundancies of GC B (GL7⁺ CXCR5⁺ B220⁺) and plasma cells (PCs, CD138⁺ B220⁻) in *Trex1*^{-/-} mice that treated with or without prasugrel. We found a significant reduction of both cells in the prasugrel-treated mice (Figure R_8).

Figure R_8. The percentage of GC B (GL7⁺ CXCR5⁺ B220⁺) and plasma cells (PCs; CD138⁺ B220⁻) in splenic cells from WT mice (n = 5) and *Trex1*^{-/-} mice (n = 5) were analyzed by FACS. Statistical analysis was performed using two-way ANOVA.

8. For comparison of the IFN-inhibitory potential of prasugrel, the authors should

also treat whole blood from SLE patients with other inhibitors of the IFN axis, such as JAK inhibitors, anifrolumab or TLR7 antagonists. Presumably, cGAS hyperactivity will only be the primary disease-causing factor in a fraction of patients. It is surprising to observe such a profound effect of prasugrel in all patients. What were the clinical characteristics of these patients? Did the authors identify any variables that could be used for patient stratification?

Response: We thank the reviewer for these valuable points and suggestions. Following these suggestions, we isolated PBMCs from peripheral blood of 10 SLE patients and treated the cells with prasugrel or JAK1/2 inhibitor ruxolitinib. Our results showed a comparable inhibiting effect of both treatments (**Fig.6c** in our revised manuscript).

Regarding the effect of prasugrel on patient cells, we re-analyzed our data in original Figure 6D and labeled the cGAMP levels for each sample. Our results showed that patients with higher cGAMP levels exhibited greater sensitivity to prasugrel treatment (**Fig.6e, f** in our revised manuscript), suggesting that plasma cGAMP levels could serve as a predictive biomarker to identify SLE patients who are likely to respond favorably to prasugrel treatment.

Given the complexity of SLE as a systemic disease, the activation of cGAS may vary across different stages of the disease. While a subset of patients tested positive for plasma cGAMP at certain timepoint, it is possible that cGAS activation may be relevant to a larger patient population. Our study suggests that SLE patients with higher cGAMP could benefit from prasugrel treatment.

Reviewer #3:

This manuscript presents interesting results that support the thesis that an anti-platelet drug, prasugrel, already used in the clinic as anti-clotting treatment, can also inhibit cGAS activation by acetylation of specific lysines. The same group published on Cell 5 years ago that cGAS activation can be inhibited through lysine acetylation by aspirin. Here the authors show that prasugrel can do what aspirin can do at a lower dose. Since the high doses of aspirin required to inhibit cGAS would have many side effects in patients, having found a different drug that carry less side effects is very significant.

The paper also shows results supporting the thesis that prasugrel can treat lupus autoimmunity, shown in two models. One model is cGAS dependent, the Trex^{-/-} mice, and one is TLR7 dependent, Balb/c mice painted on the skin with imiquimod. The inhibition by prasugrel of the autoimmunity in Trex^{-/-} mice is clear and convincing, suggesting a novel treatment in the cGAS dependent interferonopathies, like Aicardi-Gutierrez. The results of the effects of prasugrel on Imiquimod autoimmunity are also clear for the part of the splenomegaly and the expression of I-IFNs and autoantibodies, but the data on the glomerulonephritis are not strong (see below).

Moreover, the manuscript proposes that cGAS is downstream of and contributes to TLR7 activation and TLR7-dependent autoimmunity by imiquimod. The lack of some important controls weakens the rigor of the contribution of cGAS to TLR7 activation and autoimmunity.

Response: The reviewer indicated that “*this manuscript is interesting, and the inhibition by prasugrel of the autoimmunity in Trex^{-/-} mice is clear and convincing*”. We appreciate the reviewer’s encouraging comments and thank the reviewer for the constructive suggestions to improve our work. Following these suggestions, we conducted substantial experiments and revised our manuscript extensively.

In details:

1. *The authors start the manuscript with the hypothesis that cGAMP, product of cGAS activation, is higher in the plasma of SLE patients because is induced by TLR7 activation, which is high because the TLR7 expression is higher in SLE patients than healthy controls. These results could be explained in many other ways. For*

example, cGAMP and TLR7 RNA expression could correlate because higher cGAMP could be induced by some form of DNA through the activation of cGAS and STING and the production of type I-IFNs, and the latter has been shown to increase TLR7 expression. Therefore, TLR7 expression could be the effect rather than the cause of the high levels of cGAMP in SLE.

Response: We thank the reviewers for the important comments. Reviewer#1 and #2 also raised similar points. We apologize for our unclear descriptions. According to previous publications, TLR7 is proposed as a key driver of lupus in humans. For example, a functional polymorphism of TLR7 was identified predisposing to SLE in humans (*PNAS*, 2010, PMID: 20733074). The unrestricted TLR7 signaling is associated with human lupus (*Science Immunology*, 2024, PMID: 38207015). Moreover, a gain-of-function mutation in TLR7 (Y264H) was reported to cause human SLE (*Nature*, 2022, PMID: 35477763). In light of these important findings, we sought to investigate whether cGAS plays a role in TLR7-driven SLE. Therefore, we first determined whether cGAS is activated in SLE patients and then assessed the correlation between *TLR7* expression and cGAMP levels (indicator of cGAS activation) (**Fig.1b** in our revised manuscript). These results may reflect the potential interplay between cGAS and TLR7 signaling.

Using mouse models, we further showed that in the TLR7-driven SLE mice, cGAS was indeed activated (**Fig.1d, 1e** in our revised manuscript) and required (**Fig.1f-I** in our revised manuscript). We also confirmed that TLR7-stimulated IFN- α production was markedly dampened in the absence of cGAS (**Fig.1m, 1n** and **Supplementary Fig.1l, 1n** in our revised manuscript). As suggested by Reviewer#2, we used ruxolitinib, a JAK1/2 inhibitor (*The New England journal of medicine*, 2012, PMID:22375970), to block the IFN cascade downstream of TLR7. Under such condition, TLR7 agonist stimuli still effectively triggered the synthesis of cGAMP (**Fig.1o** in our revised manuscript), suggesting that the involvement of cGAS in TLR7 signaling is independent of the feedback of IFN axis. Thus, cGAS may regulate the activation of TLR7 signaling.

We further studied the role of cGAS in TLR7 signaling and found that TLR7 activation led to the increased mitochondrial permeability and the cytosol mtDNA releasement (**Supplementary Fig.2a-c** in our revised manuscript). The released mtDNAs stimulated the activation of cGAS, which is important for the effective production of IFN- α downstream TLR7 (**Fig.1c, 1m, 1n** and **Supplementary Fig.1c, 1l, 1n** in our revised manuscript). We further show that the mtDNAs are released *via* BAX and BAK macropores, as the BAK/BAX oligomerization

inhibitor MSN-125 (*Cell Chemical Biology*, 2017, PMID: 28392146) robustly suppressed the mtDNA release induced by TLR7 activation (**Supplementary Fig.2d-f** in our revised manuscript).

Thus, our new data demonstrate that cGAS is essential for the TLR7-driven IFN- α production. With these new data, we uncovered the mechanism underlying the role of cGAS in TLR7-signaling and provided important new insights into crosstalk between TLR7 and cytosolic nucleic acid sensing.

We agree with the reviewer that both *TLR7* and *cGAS* are interferon-stimulated genes (ISGs), thus the observed coordination in their expression is expected. In the revised manuscript, we rephrased our rationale for studying TLR7-driven SLE and removed data regarding the correlation between *cGAS* mRNA levels and *TLR7* expression.

2. *Fig.1c-d are important to support a direct induction of cGAS activation by TLR7 stimuli, but the legend does not clarify whether the three dots per each bar are technical replicates from one culture or from three individual cultures from three mice. More repeats of the experiments are required to support the rigor of these figures. Moreover, there is no justification for using a mix population of bone marrow precursors instead of BMDMs or BMDCs to test TLR7 responses. Furthermore, IFN α is mostly produced by plasmacytoid dendritic cells (pDCs) and therefore testing the production of IFN α in cGAS $^{-/-}$ pDCs would provide stronger data to support the dependence of the IFN α production on cGAS. Alternatively, the authors could measure IFN beta produced by BMDMs and BMDCs.*

Response: We thank the reviewer for the important suggestions to improve our work. We apologize for our unclear description. In our original manuscript, the three dots per each bar of Figure 1c-d represent the results from three independent experiments (each dot represents an individual mouse). Following the reviewer's suggestions, we repeated the experiments using six mice in each group and obtained consistent results (**Fig.1c, 1m** and **Supplementary Fig. 1c, 1l** in our revised manuscript). In the revised manuscript, we added the descriptions of experimental design in each figure legend.

In our study, we tried to examine TLR7 activation in BMDMs. The results showed that R837 or R848 treatment failed to induce *Ifna1*, *Ifna4* and *Ifnb* expressions in BMDMs (**Figure R_9**). As control, these TLR7 agonists markedly induced the expression of *Il6* and *Tnfa* in BMDMs (**Figure R_10**), which are

consistent with previous reports (*Nature*, 2022, PMID: 35477763; *The Journal of Immunology*, 2018, PMID: 30355788). Because we found that R837 stimulated IFN- α production in bone marrow cells, we used bone marrow cells in our study for TLR7 activation.

Following the reviewer's suggestion, we employed plasmacytoid dendritic cells (pDCs) to validate the role of cGAS in TLR7-driven IFN- α production. We isolated pDCs from spleen of mice using plasmacytoid dendritic cell isolation kit of Miltenyi Biotec (130-107-093). The purity of the isolated pDCs was confirmed by flow cytometry, with mouse pDC surface markers, PDCA-1 and CD11c (*The Journal of experimental medicine*, 2017, PMID: 28356390) (Supplementary Fig.1m in our revised manuscript). We then challenged pDCs from WT and *Cgas*^{-/-} mice with TLR7 agonists, R837 or R848, and measured IFN- α production. In line with our data in the manuscript, these new results demonstrated that TLR7-stimulated IFN- α production is markedly dampened in the absence of cGAS (Fig.1n and Supplementary Fig.1n in our revised manuscript).

Figure R_9. qPCR analysis of *Ifna1*, *Ifna4* and *Ifnb* expression with indicated stimuli.

Figure R_10. qPCR analysis of *Il6* and *Tnfa* expression with indicated stimuli.

3. The conclusion that R837/imiquimod induces cGAS activation and production of cGAMP is a very novel one that requires stronger results. It would be important to

provide the control of stimulating with R837 and measuring cGAMP production in cells that are both *TLR7*^{-/-} and *TLR7*^{-/-}/*cGAS*^{-/-}, to exclude the possibility that R837 induces cGAMP in a TLR7 independent manner. Moreover, would other TLR7 ligands induce the same cGAMP production or is it something specific of imiquimod? Does it require TLR7?

Response: We appreciate the reviewer's insightful comments. Following the reviewer's suggestion, we first isolated wild-type and *Tlr7*^{-/-} bone marrow cells and challenged the cells with R837 or R848, another known agonist for TLR7 (*Nature Immunology*, 2002, PMID: 11812998). We observed that R837/R848 stimuli failed to induce cGAMP production in *Tlr7* deletion cells (**Supplementary Fig.1o, p**). As the reviewer suggested, the use of *Tlr7*^{-/-}*Cgas*^{-/-} mice would be an ideal control to demonstrate the conclusion. However, given the challenge of acquiring the *Tlr7*^{-/-}*Cgas*^{-/-} mice, we adopted an alternative experimental approach. We used RU.521, a cGAS inhibitor, and M5049, a TLR7 inhibitor (*The Journal of pharmacology and experimental therapeutics* 2021, PMID: 33328334), and found that the inhibition of either cGAS or TLR7 attenuated cGAMP production stimulated by R837 or R848 (**Figure R_11**). We next treated *Tlr7*^{-/-} cells with RU.521 (cGAS inhibitor) or treated *Cgas*^{-/-} cells with M5049 (TLR7 inhibitor) and found that the blockade of both proteins with each strategy dampened the R837/R848-induced cGAMP production. (**Figure R_12**).

Together, besides R837 (Imiquimod), R848 can also stimulate cGAMP production, and this induction is TLR7-dependent.

4. *The role of cGAS in the Imiquimod-induced lupus is still unclear. Fig. 1m shows the staining for IgM in murine glomeruli as sign of lupus disease, but it is standard procedure to show the staining for IgG in the glomeruli, since IgG are the sign of a memory response against the kidney, as it is regularly observed in autoimmune mice.*

Response: We appreciate the reviewer's important suggestions. Accordingly, we performed immunofluorescence staining of IgG on kidney sections from IMQ-induced lupus mice. Consistent with previous study (*Arthritis & Rheumatology*, 2014, PMID: 24574230), we observed the deposition of IgG in the glomeruli of IMQ-induced wild-type mice at 4 weeks. Importantly, the IgG deposition was markedly attenuated by either genetic deletion of *Cgas* or by treatment with prasugrel (**Fig.1k** and **Fig.5g** in our revised manuscript).

Regarding the role of cGAS in the TLR7 signaling, we performed additional experiments and found that TLR7 activation led to the increased mitochondrial permeability and the cytosol mtDNA releasement (**Supplementary Fig.2a-c** in our revised manuscript). The released mtDNAs stimulated the activation of cGAS, which is important for the effective production of IFN- α downstream TLR7 (**Fig.1c**, **1m**, **1n** and **Supplementary Fig.1c**, **1l**, **1n** in our revised manuscript). We further show that the mtDNAs are released *via* BAX and BAK macropores, as the BAK/BAX oligomerization inhibitor MSN-125 (*Cell Chemical Biology*, 2017, PMID: 28392146) robustly suppressed the mtDNA release induced by TLR7 activation (**Supplementary Fig.2d-f** in our revised manuscript). Thus, our new data demonstrate that cGAS is essential for the TLR7-driven IFN- α production. With these new data, we uncovered the mechanism underlying the role of cGAS in TLR7 signaling and provided important new insights into the role of cGAS in the pathogenesis of TLR7-driven SLE.

5. *The histology of the Imiquimod kidney presented in Fig. In does not show the standard features of glomerulonephritis that are usually present in mice developing lupus, including the imiquimod type (PMID 35297032, 35260898). It is recommended that the H&E slides of the kidney are read by an expert in murine lupus nephritis. Same comments for Figure 5g and h.*

Response: We thank the reviewer for this valuable advice. As the reviewer indicated that our H&E slides of the kidney did not show the standard features of glomerulonephritis, we carefully read the papers mentioned by the reviewer (PMID: 35297032, 35260898). In these publications, the authors either treated NZB/NZW F1 mice with IMQ or used IMQ to treat FVB/NJcl mice. In contrast, we treated BALB/c mice with IMQ, and the standard features of glomerulonephritis are not obvious, which can be attributed to the difference of administration time and the different genetic backgrounds of mice. In the whole-kidney slides shown below, obvious inflammatory cell infiltration can be found in certain renal areas of the IMQ-induced lupus mice, which were significantly alleviated by either *Cgas* deletion or prasugrel treatment (Figure R_13).

Additionally, as suggested by the reviewer, we performed immunofluorescence staining of IgG on kidney sections from IMQ-induced lupus mice. We observed the deposition of IgG in the glomeruli of IMQ-induced lupus mice at 4 weeks. Importantly, the IgG deposition was markedly attenuated by either genetic deletion of *Cgas* or by treatment with prasugrel (Fig.1k and Fig.5g in our revised manuscript).

Following the reviewer's suggestions, all H&E-stained kidney slides (Fig.1l and Fig.5i in our revised manuscript) were reevaluated by a renal pathologist, Dr. Chunmei Wu (*Annals of the Rheumatic Disease*, 2024, PMID: 39419536), who is one of our authors in this manuscript.

Figure R_13. A. HE-stained of kidney from WT and *Cgas*^{-/-} mice treated with IMQ for 4 weeks. **B.** HE-stained of kidneys from the mice treated with IMQ for 4 weeks in indicate treatment group.

6. *The potential of prasugrel as novel therapeutic strategy in human SLE is overstated. The results in Fig. 6 indicate that prasugrel decreases the IFN Signature in cells from SLE patients. These effects are not direct indicators that prasugrel will be therapeutic in SLE since anifrolumab, which decreases the IFN Signature, has only mild effects on lupus disease. Moreover, Figure 6F reports the meta-analyses of public data that indicate that the expression of the IFN Signature inversely correlates with the cGAMP levels in the plasma, but it is not clear what is the source of the measurements of the cGAMP levels (the cited papers?) and how could serve as a predictive biomarker to identify SLE patients who are likely to respond favorably to prasugrel treatment.*

Response: The reviewer indicated that “*The potential of prasugrel as novel*

therapeutic strategy in human SLE is overstated', we agree with the reviewer that IFN signatures only serve as cellular readouts that may reflect the disease state but are not direct indicators of SLE severity. Therefore, the effects of prasugrel, as assessed by IFN signature change, require further validation in clinical trials involving SLE patients. Following this suggestion, we toned down our descriptions of conclusion statements in our manuscript. For example, in the revised manuscript, we added "potential" in the subheading of the paragraph (Line.200: Prasugrel exhibits potential therapeutic effects in SLE patient cells). We also deleted "promising" in the conclusion sentence (Line.217: Collectively, our work presents prasugrel as a ~~promising~~ therapeutic candidate for SLE).

Regarding the correlation between cGAMP levels and the inhibition of IFN signatures. In our manuscript, we showed a positive correlation between the plasma cGAMP levels and the inhibition of IFN signatures. The expression of IFN signature genes was substantially more suppressed in SLE patients exhibiting elevated plasma cGAMP levels.

We apologize for our unclear descriptions. The original Figure 6F was not a meta-analysis of public data. We analyzed our data from the cohort including 26 SLE patients that we obtained from Renji hospital, Shanghai Jiao Tong University School of Medicine. We measured the plasma cGAMP levels and examined the IFN signatures using PBMCs that treated with or without prasugrel. In the original Figure 6F, we show the trend that the expression of IFN signature genes was substantially more suppressed in SLE patients exhibiting elevated plasma cGAMP levels. The labels, such as "IFN signature (*Ref. PMID: 35477763*)" under the panels, only indicated the reference for the gene list we used. We apologize again for our unclear description and confusing labeling, and we improved the labeling and figure legends in our revised manuscript (**Fig.6c, e, f** and **Supplementary Fig.5a-f** in our revised manuscript).

7. *Methods: Statistical significance was assessed using two-tailed student's unpaired t-test even for data in which 3 or more conditions were analyzed, which would require ANOVA and post-hoc analyses.*

Response: We thank the reviewer for this important instruction. Following the reviewer's suggestion, we re-performed all relevant statistical analyses throughout our manuscript. The study designs with 3 or more conditions are now analyzed with ANOVA and post-hoc analyses (**Fig.1a, 1c-e,1g-j, 1m-o, 2i, 2k-p, 3c, 3e, 3f, 3h, 3j, 4b, 5c-h** and **Supplementary Fig.1c, 1e-h, 1l, 1n, 1p, 2a-f, 3d, 3g-i** in our

revised manuscript).

8. *The authors state that antibodies against cGAS, Acetylated-cGAS (K384+K394) and Ac-cGAS (K414) were generated in their laboratory, but it is not mentioned whether those Antibodies had been validated before and how. This validation is crucial. If it was done in a previous publication, please say so and cite the paper.*

Response: We apologize for our unclear description regarding the validation of cGAS antibodies. Antibodies against cGAS, Ac-cGAS (K384+K394) and Ac-cGAS (K414) used in this study were generated in our laboratory and were validated in our previous work (*Cell*, 2019, PMID: 30799039). We included this information in the **Methods** section of the revised manuscript (Line. 472- Line. 474 in our revised manuscript).

Minor points:

9. *Please spell out what AcK means.*

Response: Following the reviewer's suggestion, we added the full name of AcK (pan-acetyl-lysine antibody) in the revised manuscript (Line. 305).

10. *The following sentence infers a translational potential in the results of Figure 6d-e and Extended Data Fig.3c,d that goes beyond the effective potential of those data. "Because the systemic disease characteristics of SLE were unable to evaluate in patient cells, we used several published IFN signatures^{30,37-39} to reflect the autoimmune status in these cells. Significantly, prasugrel ameliorated the autoimmune status in patient cells, as indicated by the expression change of IFN signatures (Fig.6d,e and Extended Data Fig.3c,d). The mentioned figures instead show an inverse correlation between the levels of cGAMP and those of the IFN Signature. Any comment on the "autoimmune status" is unjustified and not necessary to support the relevance of the paper. Please change the wording of these sentences.*

Response: We apologize for our unclear descriptions. In our manuscript, we showed a positive correlation between the plasma cGAMP levels and the inhibition of IFN signature. The expression of IFN signature genes was substantially more suppressed in SLE patients exhibiting elevated plasma cGAMP levels.

Following the reviewer's suggestions, we replaced "autoimmune status"

with “SLE-associated immune responses” (Line. 205 – Line. 206 in our revised manuscript) according to the publications we referenced for IFN signatures (*Nature Immunology*, 2020, PMID: 32747814; *Genome Research*, 2021, PMID: 33674349; *Nature*, 2022, PMID: 35477763; *Science Immunology*, 2024, PMID: 38207055).

Point-by-Point Response:

Reviewer #3:

The authors of this paper have answered to the critiques with great effort. I especially appreciate the openness to show raw data, like the full kidney histology, and the large number of unpublished results for the reviewers' eye only. These data have dissipated past concerns. Moreover, they discovered the mechanism that link TLR7 and cGAS, making this paper very novel indeed.

Response: We thank the reviewer for the supportive comments.

Without asking new experiments, I have few suggestions to improve the clarity of the manuscript.

1. The results of the inhibitors of M5049 and Ru.521 showed in Figure R11 and R12 are very interesting and strongly support the paper. Can they be added to this manuscript?

Response: Following the reviewer's suggestion, we added these data to our revised manuscript (Supplementary Fig. 2o, p).

2. The results of R5-, R6, R7 are also very important (R7 above all) to put in perspective the conclusion of this paper with previous papers suggesting that cGAS is not important in lupus. It would be great to add them to this paper. If the authors are already planning to put them in another paper, maybe they can mention them in the discussion as data in a manuscript in preparation.

Response: We thank the reviewer for the valuable advice. Following the reviewer's suggestion, we added these data to our revised manuscript (Supplementary Fig. 2a-i).

3. Regarding the histology of the kidney in Balb/c mice treated with Imiquimod, Figure R13 shows that this model does not have the classic signs of lupus glomerulonephritis, but rather immune infiltrates in the lower tubular and medullary areas, and those infiltrates do not form with prasugrel treatment or in cgas^{-/-} mice. Therefore, figure R13 strongly supports the conclusions of the paper and would be helpful to show it. Moreover, it would remind the reader that Balb/c or in general WT mice have mild forms of kidney disease that are not exactly lupus nephritis. Showing R13 in Supplemental would NOT diminish the strength of the conclusions and would be didactic for the readers, reminding them of the limitations of the murine models with work with.

Response: We thank the reviewer for the insightful comments. Following the

reviewer's suggestion, we added these data to our revised manuscript (Supplementary Fig.1I and Supplementary Fig.6a).

4. Figure 6 supports the conclusion that prasugrel inhibits the expression of genes linked to cGAS activation and inflammation in general, and it has more effect in SLE patients with high levels of cGAMP. In Fig. 6e the general effect of prasugrel is clear with a naked eye, but the link between the levels of cGAMP and the effects of prasugrel is not that evident. After spending some time on the Figure, I am convinced but it could help the readers if the genes, rather than in alphabetical order, were clustered in functional groups, like ISGs, vs. chemokines, vs. other groups and etc, since it seems that they behave similarly inside groups, facilitating the recognition of trends and behaviors with the eye.

Response: We thank the reviewer for this important point. Accordingly, we updated **Figure.6e**, in which genes were first clustered into functional similarity groups using DAVID gene functional classification (tool) and were ordered based on these group (https://davidbioinformatics.nih.gov/gene2gene_new.jsp) (2022, *Nucleic Acids Research*, PMID: 35325185).